# Repeat-based holocentromeres of the woodrush *Luzula sylvatica* reveal insights into the evolutionary transition to holocentricity

Yennifer Mata-Sucre[1,2,9], Marie Krátká[1,3,4,9], Ludmila Oliveira[5], Pavel Neumann [5], Jiří Macas [5], Veit Schubert [6], Bruno Huettel[7], Eduard Kejnovský [3], Andreas Houben [6], Andrea Pedrosa-Harand [2], Gustavo Souza[2] & André Marques [1,8] ✉

In most studied eukaryotes, chromosomes are monocentric, with centromere activity confined to a single region. However, the rush family (Juncaceae) includes species with both monocentric (*Juncus*) and holocentric (*Luzula*) chromosomes, where centromere activity is distributed along the entire chromosome length. Here, we combine chromosome-scale genome assembly, epigenetic analysis, immuno-FISH and super-resolution microscopy to study the transition to holocentricity in *Luzula sylvatica*. We report repeat-based holocentromeres with an irregular distribution of features along the chromosomes. *Luzula sylvatica* holocentromeres are predominantly associated with two satellite DNA repeats (*Lusy1* and *Lusy2*), while CENH3 also binds satellite-free gene-poor regions. Comparative repeat analysis suggests that *Lusy1* plays a crucial role in centromere function across most *Luzula* species. Furthermore, synteny analysis between *L. sylvatica* ($n = 6$) and *Juncus effusus* ($n = 21$) suggests that holocentric chromosomes in *Luzula* could have arisen from chromosome fusions of ancestral monocentric chromosomes, accompanied by the expansion of CENH3-associated satellite repeats.

Centromeres are specialized chromosomal regions that recruit kinetochore proteins and mediate spindle microtubule binding to ensure correct chromosome segregation during mitosis and meiosis[1,2]. Most taxonomic groups have chromosomes with a size-restricted centromeric domain confined to the primary chromosome constriction,

i.e., they are monocentric[2]. However, holocentric chromosomes lack a primary constriction and exhibit molecular and epigenetic features that allow kinetochore proteins and microtubules to bind extensively along the chromosomes[2,3]. Therefore, holocentric chromosomes can tolerate large-scale rearrangements, such as chromosome fusions and

[1]Department of Chromosome Biology, Max Planck Institute for Plant Breeding Research, Cologne, Germany. [2]Laboratório de Citogenética e Evolução Vegetal, Departamento de Botânica, Centro de Biociências, Universidade Federal de Pernambuco, Recife, PE 50670-901, Brazil. [3]Department of Plant Developmental Genetics, Institute of Biophysics of the Czech Academy of Sciences, Kralovopolska 135, 61200 Brno, Czech Republic. [4]National Centre for Biomolecular Research, Faculty of Science, Masaryk University, Kamenice 5, 625 00 Brno, Czech Republic. [5]Biology Centre, Czech Academy of Sciences, Institute of Plant Molecular Biology, České Budějovice, Czech Republic. [6]Leibniz Institute of Plant Genetics and Crop Plant Research (IPK) Gatersleben, 06466 Seeland, Germany. [7]Max Planck Genome Centre, Max Planck Institute for Plant Breeding Research, Cologne, Germany. [8]Cluster of Excellence on Plant Sciences (CEPLAS), Heinrich-Heine-University, Düsseldorf, Germany. [9]These authors contributed equally: Yennifer Mata-Sucre, Marie Krátká. ✉e-mail: amarques@mpipz.mpg.de

fissions, because the rearranged chromosomes can maintain kinetochore activity, avoiding segregation problems and preserving essential genetic information during cell divisions[4–6].

Holocentric chromosomes have evolved repeatedly in animals and plants[7,8]. The lack of conclusive evidence pointing to reversions to monocentricity in any eukaryotic lineage and the sporadic distribution of holocentric versus monocentric organisms support the unidirectional transition to holocentricity[7]. Numerous evolutionary models have been proposed to explain the emergence of holocentricity from monocentric ancestors, which include alterations/loss/emergence of kinetochore genes or centromeric repetitive sequences during the process[8–11]. In the *Cuscuta* genus of the Convolvulaceae family, the transition to holocentricity was associated with not just massive changes in the kinetochore component localization on the chromosomes, but also with a loss/truncation/alteration of some important representatives of the KMN complex such as KNL2, KNL1, ZWINT1, MIS12 and NDC80[10]. The causes of the transitions, however, remain unclear, mainly because only a few holocentric species have been studied up to date and because most holocentric groups evolved a long time ago, making the factors involved in the transition difficult to determine.

In most plants, functional centromeres are epigenetically specified by the centromeric histone H3 variant (CENH3). CENH3 binding regions (hereafter CENH3 domains) in monocentric chromosomes are typically associated with extended arrays of tandemly repeated sequences (satellite DNA), which are usually highly divergent and fast evolving[12,13]. Although the role of these repeats in centromere function has not yet been fully elucidated, several possible advantages of centromeric repeats have been proposed. Satellites might have favorable monomer lengths stabilizing CENH3 nucleosome positioning or contain specific sequences, such as short dyad symmetries, forming non-B-DNA structures possibly aiding CENH3 nucleosome loading through interaction with CENH3 chaperone proteins, such as HJURP in the case of humans[1,14,15]. Nevertheless, only a few cases of holocentric species with centromeric repeats have been characterized so far. *Rhynchospora* Vahl. (Cyperaceae) holocentromeres are mainly composed of a 172-bp satellite called *Tyba*, evenly distributed along the chromosomes in ~20 kb domains and specifically colocalizing with CENH3[4,16]. In *Chionographis japonica* (Willd.) Maxim. (Melanthiaceae), several large (~2 Mb) CENH3-positive domains are associated with satellite arrays of 23- and 28-bp-long monomers[17]. Similarly, in mulberry (*Morus notabilis*) few CENH3-positive domains are associated with satellite arrays of 82-bp-long monomers[18]. In holocentric animals, known centromeric tandem repeats are even more elusive, with only the *Meloidogyne incognita* root-knot showing a 19-bp sequence box conserved within diverse centromeric satellites associated with holocentromere function[19].

Juncaceae Juss. (rushes/woodrushes), the sister family of Cyperaceae (sedges), is a cosmopolitan family comprising ~473 species[20]. *Juncus* L. (rush) and *Luzula* DC. (woodrush) represent the largest genera in the family with 332 and 124 species, respectively[20]. An interesting feature of this family is its variation in centromeric organization and chromosomal structure, making it an ideal model to address hypotheses about evolutionary processes during centromere-type transition. Although historically the entire Juncaceae family was thought to be holocentric, cytogenetic and genomic studies revealed that six different *Juncus* species are monocentric[4,21–23]. Recent chromatin immunoprecipitation sequencing revealed that *J. effusus* has repeat-based and CENH3-associated monocentromeres, consisting mainly of two tandem repeat families underlying one or up to three spaced cores of CENH3-enriched regions per chromosome[23]. On the other hand, *Luzula* species have been so far characterized as holocentric without specific centromeric repeats[3,24–26]. Although in *Luzula* species, a 178 bp satellite sharing sequence similarity with the rice centromeric repeat was

discovered[27], it is uncertain whether this satellite plays a centromeric role and the lack of a reference genome has made detailed studies of *Luzula* centromeres challenging.

Here, we perform a comprehensive (epi)genomic characterization of the chromosome-scale genome of the woodrush *Luzula sylvatica*, focusing on its holocentromere organization, repetitive fraction, and genome evolution. We show that *L. sylvatica* has a unique repeat-based centromere organization distinct from previously described holocentric species. Comparative genomic repeat profiles of 13 *Luzula* species reveal likely conservation of repeat-based holocentromeres in the genus, except for *Luzula elegans*. Further comparative genomics analysis between *Juncus effusus* and *L. sylvatica* shows footprints of extensive monocentric chromosome fusions that could have potentially played an important role in the transition to holocentricity in the lineage.

## Results

### Holocentromeres of *L. sylvatica* are predominantly repeat-based

We estimated a genome size of 1 C = 476 Mb for *L. sylvatica* (2*n* = 12) based on k-mer frequencies (Supplementary Fig. 1), and assembled a chromosome-scale reference genome sequence integrating PacBio HiFi reads and a chromatin conformation capture (Hi-C) interaction dataset available at www.darwintreeoflife.org[28] (Fig. 1 and Supplementary Fig. 1). The de novo genome assembly of *L. sylvatica* generated 1,010 contigs totaling 516.08 Mb with a GC content of 33.01%, N50 of 7.6 Mb, and Benchmarking Universal Single-Copy Orthologs (BUSCO) completeness of 93.01% (Fig. 1a–b and Supplementary Table 1). Six pseudomolecules were obtained by Hi-C scaffolding, with a total of 468.44 Mb and an N50 of 78.95 Mb (Fig. 1c and Supplementary Table 1). Similar to holocentric beak-sedges[4], as the concept of chromosome arms does not apply to holocentric species, we observed no large-scale compartmentalization or telomere-to-centromere axis, as evidenced by Hi-C contact matrix (Fig. 1c). Immunolabelling of CENH3 on *L. sylvatica* mitotic cells confirmed the holocentricity of its chromosomes (Fig. 1d). The assembly was annotated concerning major genomic sequence types, including genes, tandem repeats and transposable elements (Fig. 1e and Supplementary Fig. 2). A dispersed but structurally heterogeneous distribution of sequences along all pseudomolecules was observed, with interstitial regions highly enriched by tandem repeats and lacking genes and transposable elements (Fig. 1e and Supplementary Fig. 2).

To identify and characterize the centromeres, as well as eu- and heterochromatin regions of the *L. sylvatica* genome, we further performed chromatin immunoprecipitation followed by sequencing (ChIP-seq) for CENH3, H3K4me3, and H3K9me2, along with DNA methylation sequencing (see Methods, Fig. 1e–f). We detected 358 CENH3 domains distributed across the entire length of all chromosomes (Fig. 1e–f and Supplementary Fig. 2). Considering that one CENH3 domain is equivalent to one centromeric unit, we observed an average of 0.76 units/Mb (range 0.64–0.90 units/Mb) or 60 units (range 51-76 units) per chromosome with an average unit length of 183 kb (range 174–197 kb; Fig. 1f). Additionally, histone modification marks H3K4me3 and H3K9me2 were intermixed along the chromosomes (Fig. 1e and Supplementary Fig. 2).

The annotation of the repetitive fraction, which represents ~59% of the genome, was based on the Domain-based Annotation of Transposable Elements (DANTE), DANTE for Long terminal repeat (LTR; DANTE-LTR) and Tandem Repeat Analyzer (TAREAN) (Table 1, see Methods). Most of this fraction corresponded to satellite DNA sequences with six families representing 35.31% of the genome, where the CL1 and CL2 clusters correspond to the most abundant satellite DNAs with 25.10% and 7.06%, respectively (Supplementary Table 2 and Supplementary Fig. 3). CL1 is a 124-bp satellite, named hereafter as *Lusy1* (Supplementary Table 2 and Supplementary

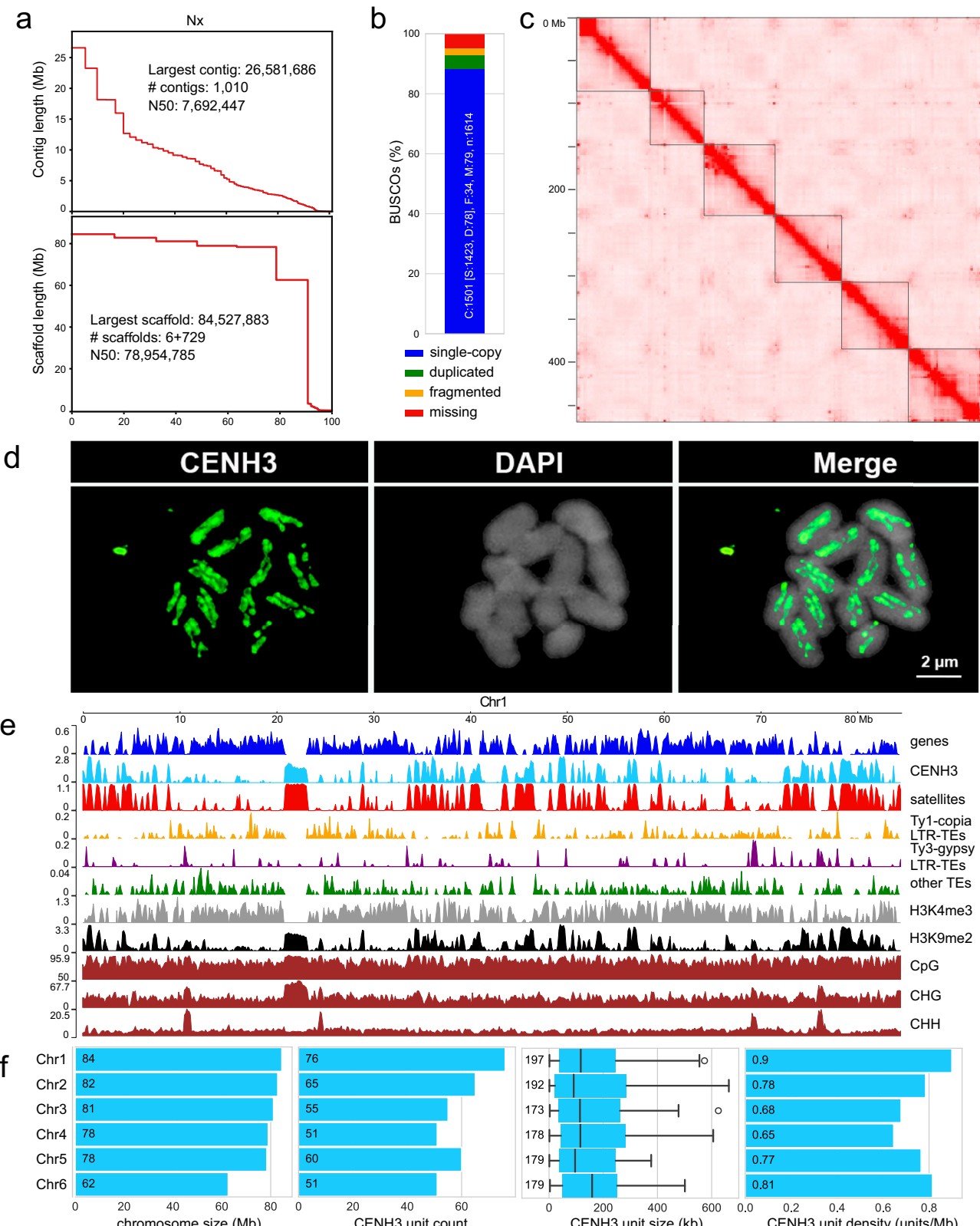

Fig. 3b). CL2 is a satellite consisting of two variants of 174 and 175 bp sharing 62% similarity (hereafter referred to as *Lusy2*; Supplementary Table 2 and Supplementary Fig. 3), and only 30% similarity to *Lusy1*. The other five satellite DNAs have monomers with 31 to 182 bp, amounting to less than ~2% in the genome each (Fig. 2a and Supplementary Table 2). Retrotransposon elements were less abundant than satellites, making up 18% of the genome (Table 1). LTR

retrotransposons of the *Ty1-copia* superfamily were the most represented with the *Angela* lineage being the most abundant (10.15%; Table 1 and Fig. 1d).

Satellite DNA families' distribution varied across the genome (Fig. 2a). *Lusy1* was spread throughout all pseudochromosomes, with higher densities in interstitial regions (Fig. 2a). In contrast, the other satellites (LsylSAT3–6) were found preferentially near telomeres or

**Fig. 1 | Genome assembly and annotation of holocentric *L. sylvatica*. a** Statistics of the *L. sylvatica* genome assembly and the final scaffolding. **b** BUSCO assessment for completeness of genic space with the viridiplantae_odb10 dataset, using the entire genome assembly. **c** Intra and inter-chromosome contact matrices of *L. sylvatica*. Color intensity represents contact frequency. Dark lines mark chromosomal boundaries. Boxes along the diagonal represent interactions within the same chromosome (cis), as expected for holocentric chromosomes. **d** Immunostaining of *L. sylvatica* holocentromeres using anti-CENH3 antibody (green). **e** *L. sylvatica* Chromosome 1 detailed view showing the dispersed density distribution of main genomic features: genes, CENH3, satellite DNA, LTR *Ty1-copia*, LTR *Ty3-gypsy*, H3K4me3 and H3K9me2 histone marks, and DNA methylation (CpG, CHG and

CHH); as typical for holocentric chromosomes. Bin sizes of 100 kb. The distribution of features on all chromosomes is reported in Supplementary Fig. 2. **f** Plots of chromosome size, number of discrete CENH3 domains (units), their size (<700 kb shown), and density (units/Mb). The CENH3 unit size boxplot follows the definition in seaborn data visualization package where central lines represent median value, boxes represent 1st and 3rd quartiles, and whiskers represent the data range without outliers, defined as observations further than 1.5 of interquartile range from the respective (1st or 3rd) quartile. The number of observations (units) on each chromosome corresponds to the CENH3 unit count panel. Source data are provided as a Source Data file.

---

irregularly distributed on the chromosomes (Fig. 2a and Supplementary Table 2). We localized in situ the two most abundant putative repeats (*Lusy1* and *Lusy2*) to corroborate the pattern obtained in silico. We observed that *Lusy1* shows a line-like distribution across the entire length of each sister chromatid (Fig. 2b), in a similar pattern to other holocentromeric repeats[16]. *Lusy2*, while also present on all chromosomes, shows a more diffuse and dispersed pattern. *Lusy2* appears to be present in areas where *Lusy1* is less enriched, suggesting a complementary distribution between the two satellites (Fig. 2b). Furthermore, in interphase nuclei, *Lusy1* signals are more focused compared to more dispersed *Lusy2* signals, with clear occurrences of co-localization as well as regions where *Lusy1* and *Lusy2* signals do not overlap (Fig. 2b). These results suggest that although *Lusy1* and *Lusy2* may occupy shared regions, they also maintain distinct territories within chromatin, further supporting the idea of their distinct roles in chromosomal organization.

## Table 1 | Genome proportion (in %) of the repetitive sequences in the genome assembly of *Luzula sylvatica* by using DANTE-LTR and TAREAN

| Element | Total length | Proportion |
|---|---|---|
| LTR | | |
| Ty1-copia/Angela | 47,526,670 | 10.15 |
| Ty3-gypsy/non-chromovirus/Athila | 12,206,459 | 2.61 |
| Ty1-copia/SIRE | 7,396,917 | 1.58 |
| Ty1-copia/Ivana | 4,937,954 | 1.05 |
| Ty1-copia/Bianca | 3,399,411 | 0.73 |
| Ty1-copia/Ale | 1,844,668 | 0.39 |
| Ty1-copia/Tork | 1,567,483 | 0.33 |
| Ty3-gypsy/chromovirus/Reina | 1,001,818 | 0.21 |
| Ty3-gypsy/chromovirus/Tekay | 693,833 | 0.15 |
| Ty1-copia/Ikeros | 689,905 | 0.15 |
| Ty1-copia/TAR | 481,869 | 0.10 |
| Ty1-copia/Alesia | 358,026 | 0.08 |
| Ty3-gypsy/chromovirus/CRM | 176,493 | 0.04 |
| non-LTR | | |
| LINE | 2,437,499 | 0.52 |
| Pararetrovirus | 43,979 | 0.01 |
| Class_II-TIR | 568,495 | 0.12 |
| Class_II-Helitron | 89,950 | 0.02 |
| Tandem repeats | | |
| Satellites | | 35.31 |
| rDNAs | | 1.43 |
| Low_complexity | 2,072,497 | 0.44 |
| Simple_repeat | 17,294,902 | 3.69 |
| Unknown | 4,273,627 | 0.01 |
| Total | | 59.11 |

To investigate whether *L. sylvatica* holocentromeres are repeat-based, we performed a comparison analysis of these satellites with CENH3 ChIP-seq data. ChIP-seq showed CENH3 enrichment for *Lusy1* and *Lusy2* repeats and depletion in LTR transposable elements throughout the *L. sylvatica* genome (Fig. 2c). DNA methylation was similar between *Lusy1/2* sequences, being highly enriched in CpG and CHG contexts at levels comparable to those of TEs. Regulatory sequences flanking the transcribed region of genes were depleted of CpG methylation compared to intergenic regions and centers of the gene bodies (Fig. 2c). As recently reported for *Rhynchospora*[4], CHG (but also CHH) methylation seems to increase toward the borders of *Lusy* satellites (Fig. 2c), reinforcing the idea of an evolutionary conserved epigenetic regulation of repeat-based holocentromeres in the cyperid clade. By overlapping the annotation of the *Lusy1* and *Lusy2* centromeric repeats with CENH3 domains, we observed that centromeric units are mainly composed of *Lusy1* (*n* = 232 out of 358 domains) and/or *Lusy2* sequences (*n* = 96) (Fig. 2d–e and Supplementary Fig. 4). Additionally, we found a small subset of CENH3 domains (*n* = 33) associated with satellite-free regions, which were mainly composed of low-complexity repeats (54%) and LTR-TEs (26%) (Fig. 2e and Supplementary Fig. 5). Although satellite-free CENH3 domains were depleted of genes, they often contain transposable elements (16 out of 33 domains; Fig. 2e). *Athila* elements belonging to *Ty3-gypsy* family were the most abundant, making up nearly 18% of the length of the satellite-free CENH3 domains while representing only ~3% of the genome (Supplementary Table 3). Satellite-free CENH3 domains were also positively correlated with CpG, CHG, and CHH methylation, similar to *Ty3-gypsy* TEs (Fig. 2f and Supplementary Fig. 6), suggesting that these two features characterize the same genomic niche. The presence of satellite-associated CENH3 domains was further confirmed by immunostaining followed by fluorescent in situ hybridization (Immuno-FISH) analyses, where satellite *Lusy1* signals partially colocalize with CENH3 domains along the chromosome (Fig. 3a–b). Therefore, *L. sylvatica* represents a case of repeat-based holocentromeres that are mostly, but not exclusively, composed of *Lusy1* repeats.

Using in silico mapping data, we also identified arrays of satellites *Lusy1* and *Lusy2* that lack association with CENH3 (hereafter referred to as nonfunctional). For *Lusy1*, 247 out of 704 arrays (35%) overlap with CENH3 domains (functional). The length of these overlapping regions was 47 Mb out of 77 Mb in total (60%). For *Lusy2*, 107 out of 952 arrays (11%) contained CENH3 domains, making up 10 Mb out of 43 Mb (24%) of total length. Nonfunctional arrays tended to be smaller than the functional arrays of the same satellite family, with an average length of functional/nonfunctional arrays of 189 kb/20 kb and 94 kb/20 kb for *Lusy1* and *Lusy2*, respectively (Fig. 3c). Functional *Lusy1* arrays contained a higher abundance of dyad symmetries. In *Lusy2*, this difference was not significant (Fig. 3d). Functional and nonfunctional arrays also differ in their inter-array sequence similarity. Functional arrays of both *Lusy1* and *Lusy2* satellites had higher average similarity across discrete arrays compared to nonfunctional arrays (88.0 vs. 87.2% and 89.6 vs. 85.1% for *Lusy1* and *Lusy2*, respectively; Fig. 3e). Interestingly,

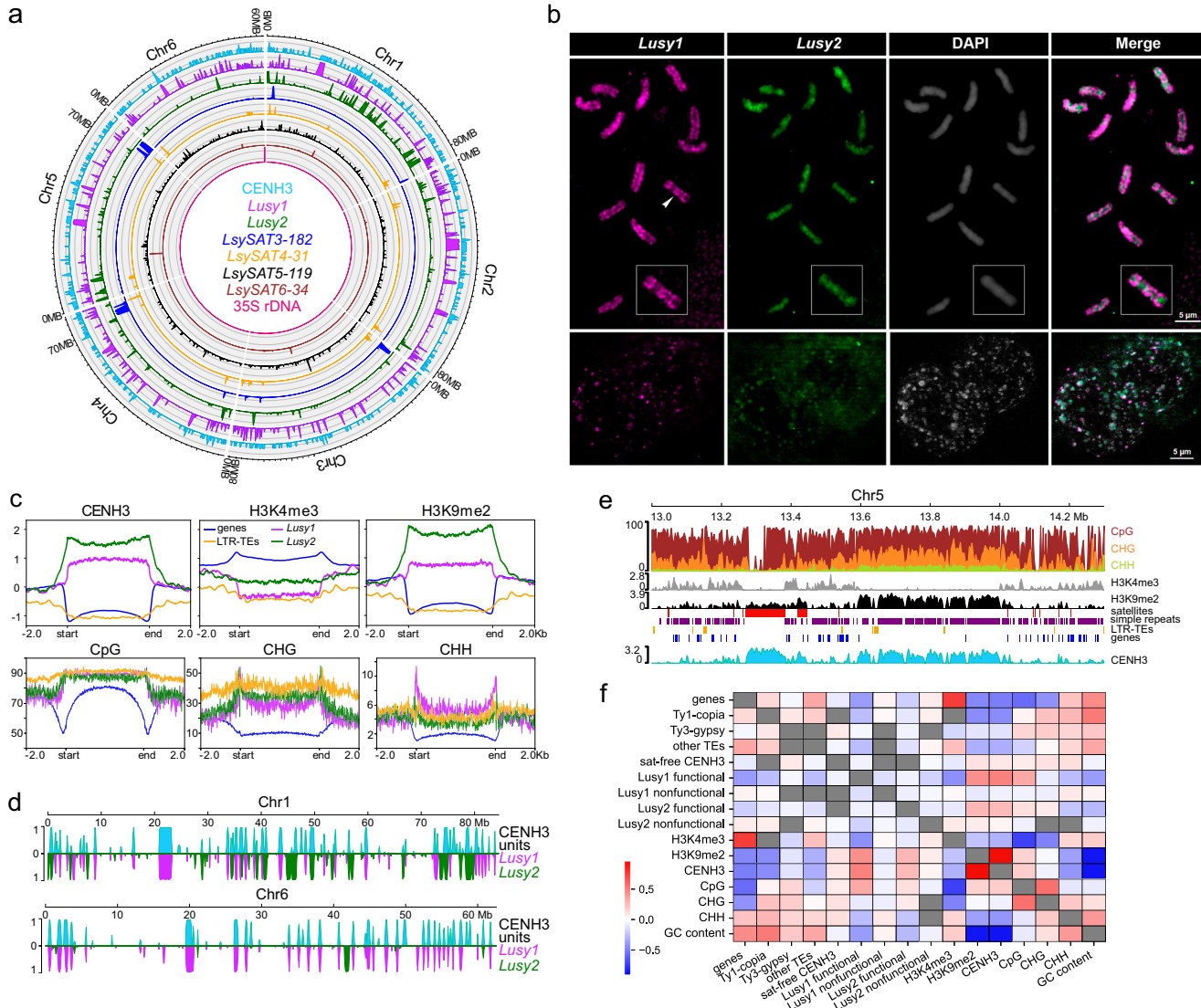

**Fig. 2 | _L. sylvatica_ holocentromeres are based on satellite repeats. a** Circos distribution of the main classes of tandem repeats and CENH3 domains with a 300 kb sliding window. **b** Fluorescent in situ hybridization (FISH) showing discontinuous linear-like spreading of the _Lusy1_ (magenta) and more disperse pattern of _Lusy2_ (green) repeats in the nucleus ($n = 10$) and metaphase chromosomes ($n = 10$). An arrowhead points to the chromosome enlarged in the inset. **c** Metaplots showing the enrichment of CENH3, H3K4me3, H3K9me2, CpG, CHH, and CGH from the start and end of different types of sequences: genes (blue), LTR transposable elements (yellow), _Lusy1_ (magenta) and _Lusy2_ repeats (green). ChIPseq signals are shown as log2 (normalized RPKM ChIP/input). Methylation signals are shown as a

percentage of methylated bases in each (CpG, CHG, CHH) context. **d** Proportion of CENH3 domains (light blue), _Lusy1_ (magenta) and _Lusy2_ (green) arrays in 100 kb windows. Distribution on all chromosomes is reported in Supplementary Fig. 4. **e** Close-up view of a genomic locus showing both _Lusy_ satellite-based and satellite-free CENH3 domains. Distribution on all chromosomes is reported in Supplementary Fig. 5. **f** Correlogram of genomic features in 100 kb windows ($n = 4694$). Gray fields indicate values on the diagonal and non-significant values of the Spearman coefficient after multiple-testing correction (see Methods). Source data are provided as a Source Data file.

nonfunctional _Lusy1_ arrays show a clear bimodal distribution with one of the groups having a higher similarity than the corresponding functional arrays (Fig. 3e). Epigenetic status of the functional array chromatin also shows a striking contrast, since functional centromeric regions (i.e., _Lusy1_ and _Lusy2_ functional arrays, satellite-free centromeric units) are enriched with heterochromatin mark H3K9me2 and depleted of euchromatin mark H3K4me3, while the nonfunctional arrays are the opposite (Fig. 2f and Supplementary Fig. 6b).

## Conservation of KNL1 and NDC80 kinetochore proteins
In _Cuscuta_, the transition to holocentricity was associated with massive changes in the localization of CENH3 and the kinetochore proteins KNL1, MIS12, and NDC80, representing the three complexes of the

KMN network[10,29]. Unlike _Cuscuta_ species, _Luzula_ still possesses centromeric activity associated with CENH3 (Fig. 1d and Supplementary Fig. 7a)[24,25]. To test whether the kinetochore assembles along the poleward chromosome surface, as expected for holocentric chromosomes, we examined the localization of KNL1 and NDC80 in two holocentric woodrushes, _L. sylvatica_ and _Luzula nivea_ as well as in the related monocentric common rush _J. effusus_. Antibodies against KNL1 and CENH3 revealed a co-localized distribution in _L. sylvatica_ metaphases demonstrating that KNL1 functionally integrates with CENH3 at centromeres during cell division (Fig. 4a). Furthermore, KNL1 showed a similar pattern in both _L. sylvatica_ and _L. nivea_, with signals detected as multiple clusters along the poleward surface of chromosomes, where microtubules attach (Fig. 4b and Supplementary Fig. 7a and 8; Supplementary Movie 1 and 2). In addition, immuno-FISH signals from the

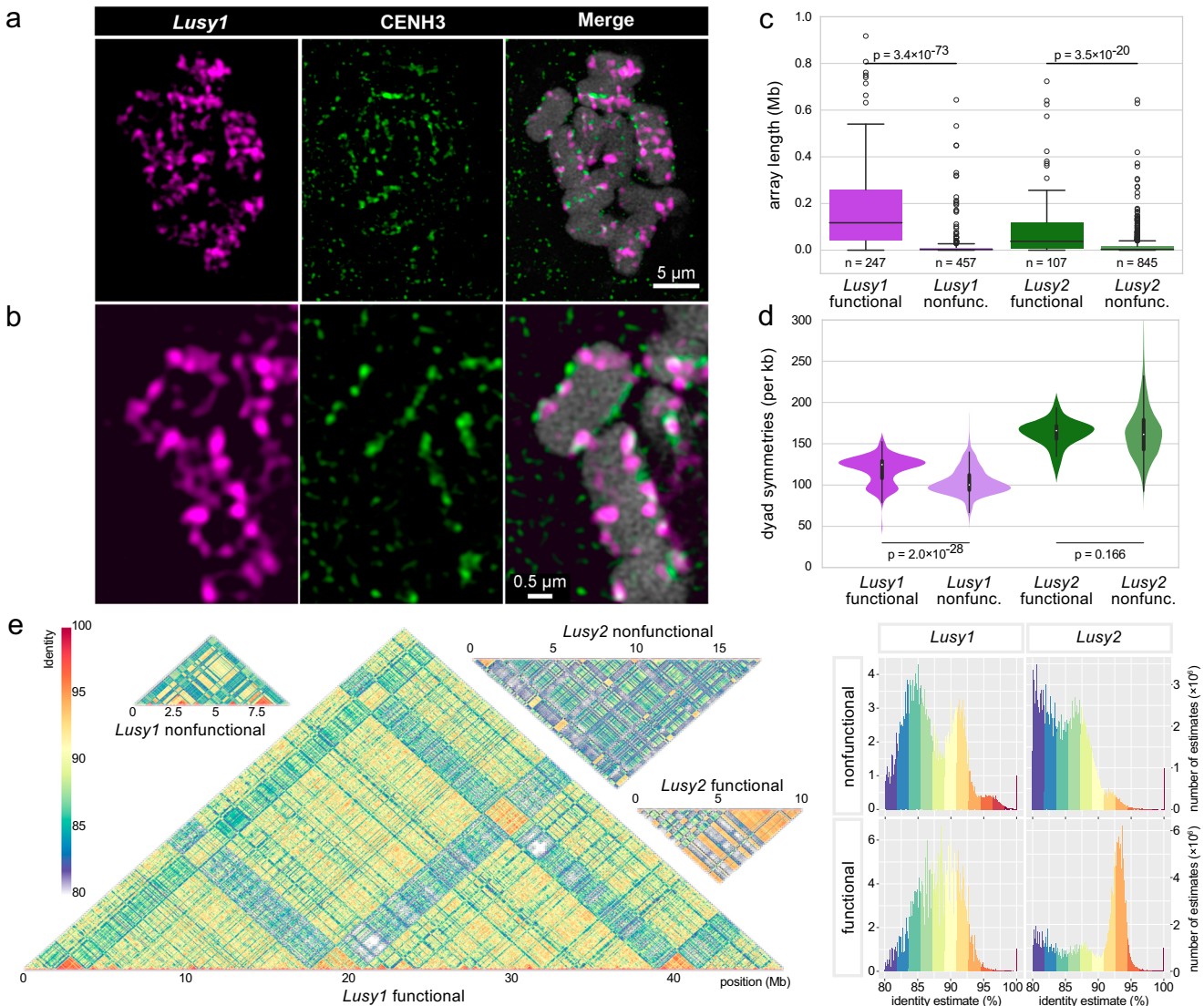

**Fig. 3 | Association of CENH3 with centromeric satellite arrays. a–b** Immuno-FISH showing partial colocalization of *Lusy1* repeats (magenta) and CENH3 (green) in metaphase chromosomes (counterstained with DAPI, gray). **c** Box plot of sizes of centromeric satellites *Lusy1* and *Lusy2* arrays associated with CENH3 (functional) or non-centromeric (nonfunctional). Statistical significance was tested using two-tailed Mann-Whitney U test. The number of observations in each group (array count) is indicated below individual boxes. **d** Abundance of dyad symmetries in functional and nonfunctional arrays of centromeric satellites. Statistical significance was tested using one-tailed Mann-Whitney U test. The number of observations (array counts) is identical to Fig. 1, panel **c**. **e** Homogeneity of functional array fragments (regions overlapping CENH3 domains) and whole nonfunctional

arrays of centromeric satellites *Lusy1* and *Lusy2*. Dot plots show sequence similarity between groups of concatenated arrays from the entire genome (left), histograms show the frequency distribution of similarity values (right). Images in **a** and **b** represent single slices of 3D-SIM image stacks. Dot plots are shown proportional to their genomic abundance. Boxplots (panel **c**) and inner boxes of violin plots (panel **d**) follow the definition in seaborn data visualization package where central lines (points in violin plot) represent median value, boxes represent 1st and 3rd quartiles, and whiskers represent the data range without outliers, defined as observations further than 1.5 of interquartile range from the respective (1st or 3rd) quartile. Source data are provided as a Source Data file.

centromeric repeat *Lusy1* presented partial overlap with KNL1, where *Lusy1* in a clustered pattern is contrasting the more continuous lines observed for KNL1 where microtubules attach (Fig. 4c). Although there is some centromeric association of *Lusy1*, this repeat is not bound exclusively to the centromere as observed at the genomic level (Fig. 4c). Unlike the KNL1 protein, NDC80 signals were observed only in *L. nivea* (Supplementary Fig. 7a; Supplementary Movie 3). Absence of NDC80 immunosignals in *L. sylvatica* could be due either to low amino acid sequence similarity with the target sequence developed in *Cuscuta*, or due to sensitivity of the protein during the cell fixation process, as discussed by Oliveira et al.[29]. In *J. effusus*, KNL1 and NDC80 showed a specific dot-like localization in the primary constriction region of the chromosome, also associated with

microtubule attachment sites (Supplementary Fig. 7b; Supplementary Movie 4 and 5). KNL1 and NDC80 associate with spindle-binding sites detected by antibodies against α-tubulin, indicating that both proteins have a conserved kinetochore function in *Luzula* and *Juncus*.

**Lusy1 and Lusy2 satellites are present across the genus Luzula**
To determine whether repeat-holocentromeres are conserved in other species of the genus *Luzula*, both an individual and comparative analysis of the repeatome using RepeatExplorer2 was performed in 13 species, including *L. sylvatica* (Supplementary Table 4; Supplementary Data 1 and 2). The global genomic proportion of repetitive DNA varied from 35.24% (*Luzula pilosa*) to 66.29% (*Luzula wahlenbergii*) (Supplementary Data 1). In general, satellites were the most abundant class of

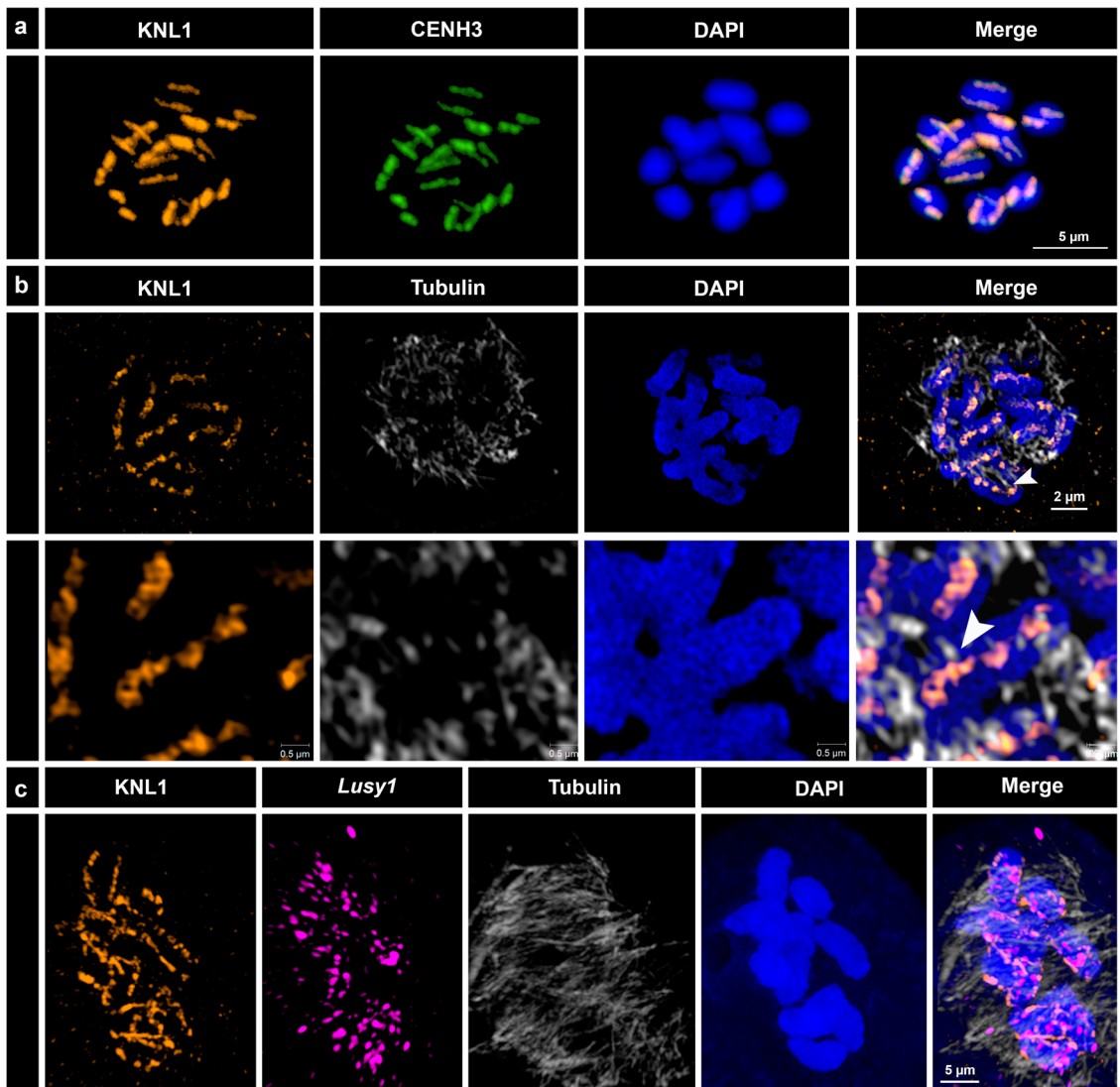

**Fig. 4 | Detection of α-tubulin and kinetochore proteins in metaphase chromosomes of *L. sylvatica*. a** KNL1 (orange) and CENH3 (green) proteins co-localize and show a holocentric distribution (*n* = 15). **b** KNL1 protein localizes specifically to the centromere surface, where microtubules (gray) bind (*n* = 15). **c** Co-detection of KNL1, *Lusy1* repeats (purple), and α-tubulin (*n* = 15). It is noteworthy that at this stage, some microtubule ends already colocalize with KNL1 proteins (indicated by arrowheads, see Supplementary Figs. 7 and 8). Maximum intensity projections of 3D-SIM image stacks.

repeats, comprising up to 49% of the *Luzula sudetica* genome. The centromeric satellite *Lusy1* was one of the most abundant among all satellites, representing up to 47.41% of *L. sudetica* genome but only 3.33% of the *L. nivea* genome and entirely absent in *Luzula elegans* (Supplementary Data 1). *Lusy2* also showed variation in abundance among species, ranging from 0.34% (*Luzula multiflora* subsp. *frigida*) to 31.03% (*Luzula luzuloides*), being also found in *L. elegans* genome (0.72%; Supplementary Data 1), a species with previously undetected holocentromeric repeats[3]. LTR retrotransposons revealed variable abundances among species, with the *Ty1-copia* superfamily being the most represented (1.21% in *L. pilosa* to 41.35% in *L. elegans*; Supplementary Data 1).

Comparative repeat analysis resulted in 166 shared clusters (Fig. 5a and Supplementary Fig. 9; Supplementary Data 2). Variants of *Lusy1*, the most abundant satellite family in *Luzula*, were found in all analyzed species, except in *L. elegans*, where this satellite was not detected even in an additional fine search of the raw sequencing reads (Fig. 5a). Different variants of the *Ty1-copia Angela* lineage were found in high abundance among the species, being more dominant in the genomes of *Luzula arcuata* and *L. elegans* (Fig. 5a; Supplementary

Data 2). *Ivana* and *SIRE Ty1-copia* lineages were also shared among all species, although they exhibited lower abundance than *Angela* (Supplementary Data 1 and 2).

Because *Lusy1* and *Lusy2* were the most abundant satellites in the comparative analysis, consistent with the observation from the *L. sylvatica* genome, we performed FISH to confirm their distribution also in *L. nivea*, the species with very low abundance of *Lusy1*. Like *L. sylvatica*, the FISH signals of *Lusy1* in *L. nivea* showed a line-like distribution along the chromosomes. However, exhibiting both enriched and depleted labeled chromosomal regions. Furthermore, *Lusy2* showed clustered signals enriched at interstitial and terminal regions in a non-linear pattern (Fig. 5b). These results suggest a similar repeat-based holocentromere organization for other *Luzula* species as well.

**Chromosome fusions drive karyotypic evolution in *Luzula***

Chromosomes from some grasses and several holocentric species have undergone extensive karyotypic rearrangements through fusions[4,5,30]. To investigate the possible association between holocentricity and chromosome fusions, we analyzed synteny between the

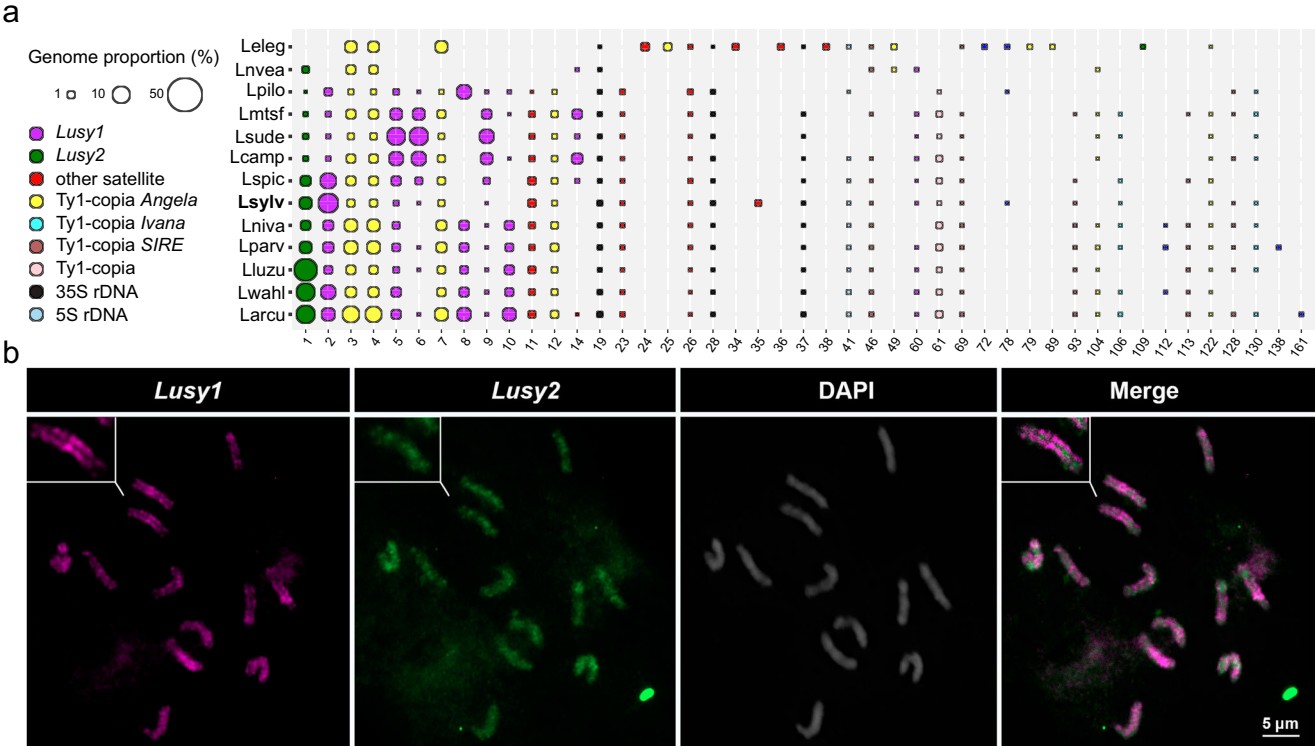

**Fig. 5 | Comparative analyses of the main types of repetitive sequences in *Luzula* species. a** Comparative analyses of the abundance of the main types of repetitive sequences in *Luzula* species. Code names correspond to Larcu: *Luzula arcuata*, Lcamp: *Luzula campestris*, Leleg: *Luzula elegans*, Lluzu: *Luzula luzuloides*, Lmtsf: *Luzula multiflora* subsp. *frigida*, Lniva: *Luzula nivalis*, Lnvea: *Luzula nivea*, Lparv: *Luzula parviflora*, Lpilo: *Luzula pilosa*, Lspic: *Luzula spicata*, Lsude: *Luzula sudetica*, Lsylv: *Luzula sylvatica*, Lwahl: *Luzula wahlenbergii*. The size of the ball is proportional to the genome abundance of that cluster for each species. The colors of the balls correspond to different repetitive sequence types (see Supplementary Table 4 for details). **b** FISH showing a wide-spreading localization of satellite *Lusy1* (magenta) and a dispersed pattern of *Lusy2* (green) repeats in mitotic metaphase chromosomes of *L. nivea* (N = 10). Chromosomes were counterstained with DAPI. Source data are provided as a Source Data file.

genomes of the holocentric *L. sylvatica* and the monocentric *J. effusus* species (Fig. 6 and Supplementary Fig. 10). Considering $n = 20$ as the putative ancestral karyotype for the family Juncaceae[31], the synteny analysis between the two genomes revealed that the chromosomes of *L. sylvatica* consist of fused blocks from *J. effusus* chromosomes (dysploid with $n = 21$), resulting in a descending dysploidy to $n = 6$ (Fig. 6a). Despite their high chromosome number and centromere-type differences, small arrangements and large syntenic blocks were identified between both genomes, indicating well conserved genomic structures in this family (Fig. 6b and Supplementary Fig. 10; Supplementary Data 3). A total of 86.4% (23,016 gene pairs) of the *J. effusus* genome is syntenic with *L. sylvatica*. Within this fraction, fine-scale synteny analysis revealed several large centromeric units that appear to be conserved between *J. effusus* and *L. sylvatica* genomes, despite the extensive variation of centromere locations and the divergence of >60 Myr of these genomes (Supplementary Figs. 11 and 12). This evidence of fusions and chromosomal rearrangements was also found using synteny of individual (unscaffolded) contigs of *L. sylvatica* and *J. effusus* (Supplementary Fig. 13).

We have recently shown that the holocentromeric repeat *Tyba* can be involved in facilitating end-to-end chromosome fusions in *Rhynchospora* species[4]. To assess the possible role of *Lusy1* and *Lusy2* in the fusion regions observed in *Luzula* genomes, we looked for specific enrichment of these repeats and telomeric repeats at the fusion regions. We found evidence for interstitial telomeric sites (ITS) using FISH experiments (Fig. 6c). Looking at the telomere annotation of *L. sylvatica* genome assembly, we observe three instances of possible interstitial telomeric sites localized in or near a fusion region (Supplementary Fig. 14). Further, the size of the fusion regions in *L.*

*sylvatica*, defined as the space between syntenic blocks, revealed a size range from 10 kb to 8 Mb. At the block boundaries (flanking regions of 50 kb), we observed a positive association with genes (Fig. 6d and Supplementary Fig. 15). Large fusion regions (>100 kb) also contained satellites and/or transposable elements (Fig. 6e). However, the enrichment of the large fusion regions with repeats is not prominent enough to be recognized at the scale of genome-wide colocalization between features (except possible enrichment for functional *Lusy1* arrays; Fig. 6d).

## Discussion

Holocentromeres have evolved from a monocentric ancestor multiple times during the evolution of eukaryotes, and despite the convergent appearance of extended centromere, each of these events results in specific genome organization and adaptation of the kinetochore protein machinery[4,8,11,32]. In *Cuscuta*, holocentricity co-occurs with CENH3 independent mitotic spindle attachments and extensive changes in kinetochore structural and regulatory protein genes[10,29]. In insects, transitions to holocentricity are associated with the loss of CENH3, while the inner kinetochore complex remained relatively conserved[9,33]. In cases where the kinetochore proteins' function is maintained, their localization can present a spectrum between continuous line-like and discrete cluster-like distribution along the chromosome[2]. While previously studied *Luzula* species (*L. elegans* and *L. nivea*) display a line-like distribution of CENH3[24,25], we detected signals more resembling a cluster-like distribution for CENH3 and KNL1 proteins on chromosomes in a less decondensed state in *L. sylvatica*, suggesting a more discontinuous and dynamic holocentromere organization in this species.

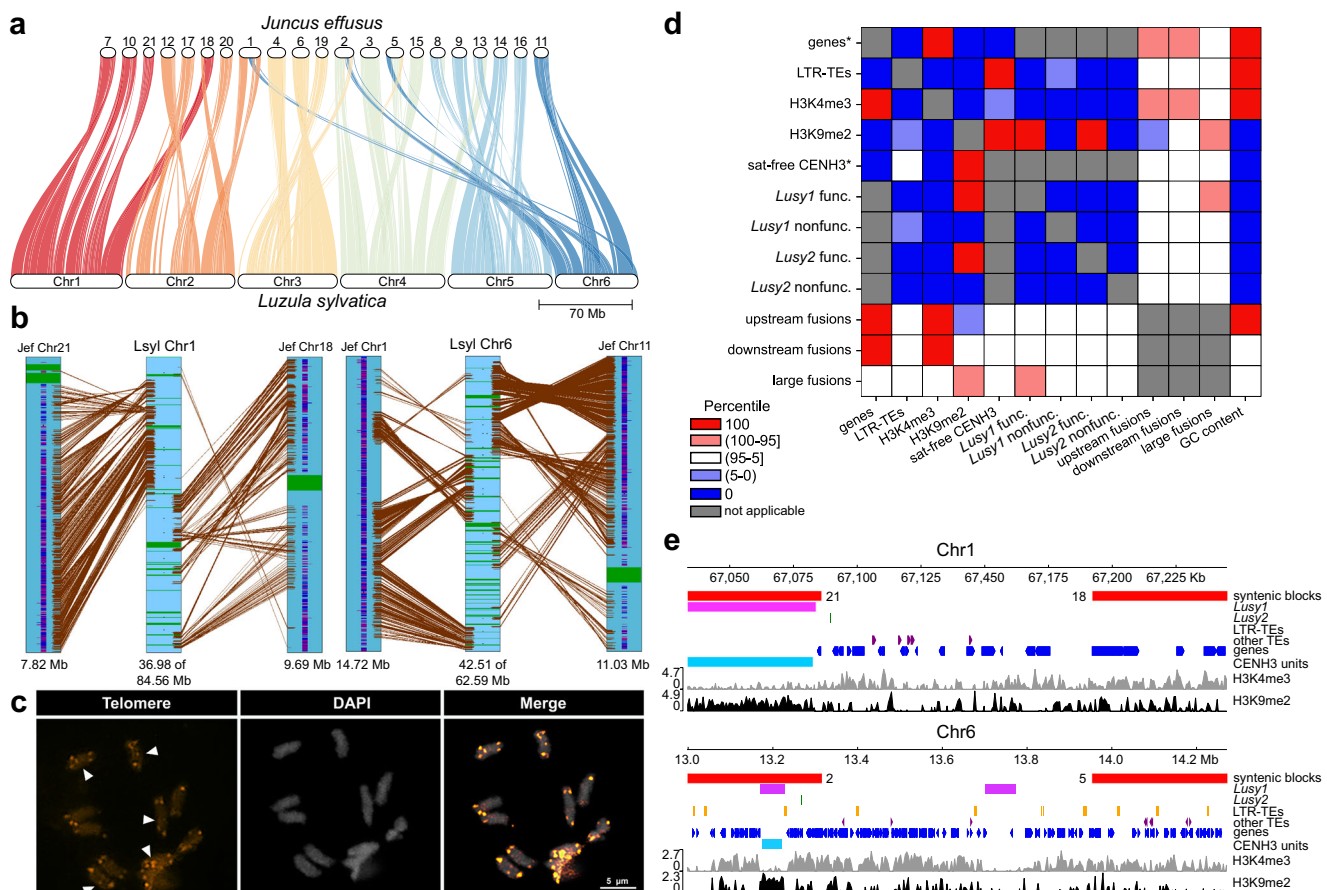

**Fig. 6 | Genome synteny comparison of *L. sylvatica* and *J. effusus*. a** Genome synteny patterns showing macro-conserved blocks of *J. effusus* that were fused into *Luzula* chromosomes. **b** *L. sylvatica* chromosome 1 and 6 (Lsyl) showing the fusion of syntenic blocks of *J. effusus* (Jef) 1, 11, 18 and 21. Genes, CENH3 and telomere domains are annotated as blue-purple, green and black stripes, respectively. **c** FISH of telomeric DNA showing chromosomes with remnant interstitial telomeric sites, suggesting ancestral fusion events (*n* = 5). **d** Colocalization between major genomic and epigenomic features and fusion regions (columns) based on comparison of overlap with simulated regions (rows). Heatmap values show the rank of real overlap values among a distribution of overlaps with simulated regions. Asterisk points to exclusion of satellite array positions from permitted simulated region locations, for the rest of the features, the whole genome was used for simulated regions. **e** Examples of fusion sites on Chr1 and Chr6 in *L. sylvatica*. Source data are provided as a Source Data file.

We show that functional holocentromeres in *L. sylvatica* are mainly made of *Lusy1* and partially *Lusy2* satellite repeats organized as kilobase-scale, non-uniformly spaced CENH3-positive centromeric units. However, the presence of several repeat-less centromere units suggests a more complex determination of centromere function in this species that needs to be further examined. These units display heterochromatin-typical characteristics and alternate with euchromatin, rich in coding regions, along the chromosome. Previous studies have identified a potential 178 bp centromeric tandem repeat in *L. nivea* and other *Luzula* species with similarity to the centromeric satellite *RCS2* from rice[27]. Indeed, the 178 bp satellite shares 87% similarity with the *Lusy2* satellite, which we found in all species. *Lusy2* partially enriches centromeric regions but does not cover entire chromosomes like 124 bp *Lusy1*, which encompasses the holocentromere of *L. sylvatica* and, along with its abundance in most analyzed *Luzula* species, suggest that *Lusy1* is the primary centromeric satellite. An exception is the early diverging species *L. elegans*, where *Lusy1* is absent and *Lusy2* only represents <1% of the genome. None of the 20 previously analyzed satellite repeats in *L. elegans* exhibited a centromeric pattern, despite repetitive sequences making up ~60% of the genome[3,34]. Repeat-based holocentromeres have been previously reported in *Chionographis japonica*[17], *Rhynchospora*[4,16,35], *Eleocharis*[36,37], in the nematode *Meloidogyne incognita*[19], and in mulberry[18]. *Luzula elegans* lacks *Lusy1*, mirroring the absence of the holocentromeric repeat *Tyba* in early diverging *Rhynchospora* lineages, which instead have diverse satellite arrays arranged in block-like patterns[38,39]. In both of these genera, the colonization of holocentromeres by contemporary genus-specific centromeric satellite families occurred only after the process of transition to holocentricity began[39].

Our results raise questions about the expansion and functional role of satellites in holocentromeres. A process similar to the establishment of neocentromeres as found in monocentrics could be taking place. Neocentromeres can arise in heterogeneous genomic regions that become subject to rapid cycles of invasion and purification of repetitive sequences through satellite homogenization[40]. In *L. sylvatica*, we have identified several satellite-free centromeric units reminiscent of maize de novo centromeres[41] in their gene-poor region targeting, CHG and CHH methylation, and possible association with *Ty3-gypsy* elements. These units could then become a subject of competition for centromere dominance between *Lusy* satellites driven by satellite homogenization and evolutionary selection pressure for centromere stabilizing effects such as advantages in CENH3 loading[15] or nucleosome formation and positioning[1]. An analogous process of acquisition of heterochromatin epigenetic modifications, accumulation of transposable elements, and invasion of satellite repeats has been described in monocentrics as evolutionary new centromere maturation[42,43].

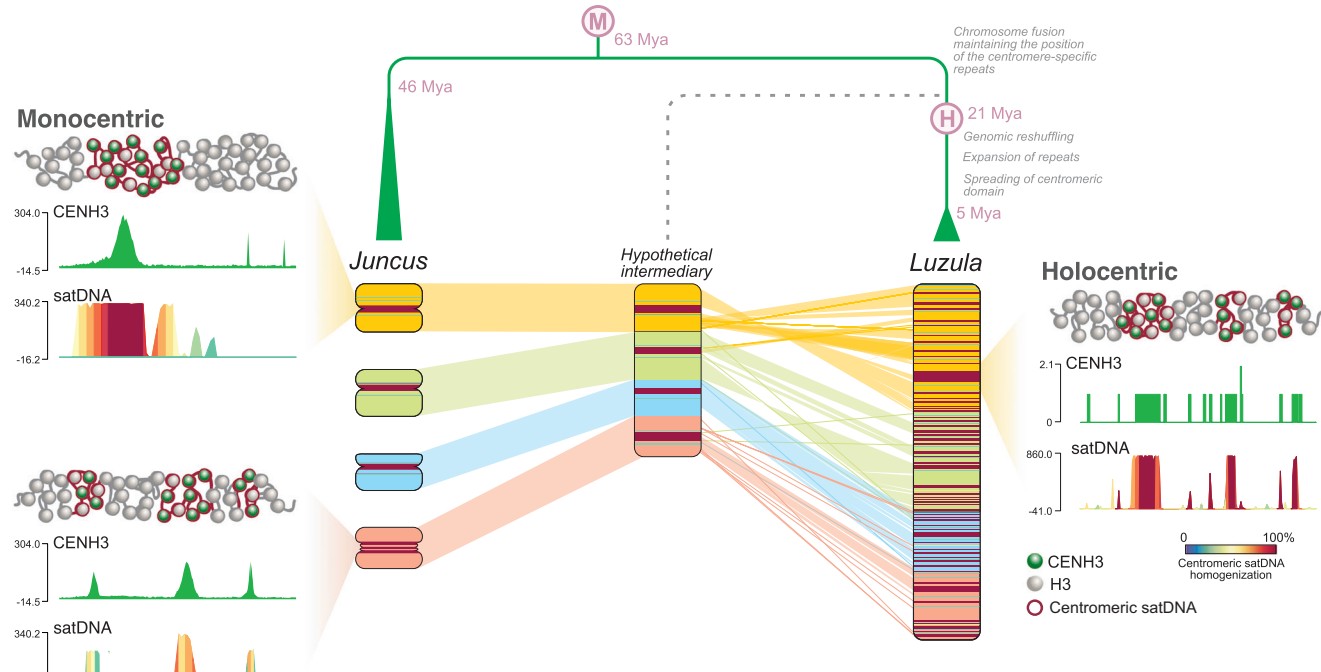

**Fig. 7 | Hypothetical model for the origin of holocentricity in woodrushes.** After several fusions of whole atypical monocentric chromosomes (*Juncus*-like type), centromeric domains were initially conserved in the larger chromosomes (hypothetical intermediate state), forming polycentric chromosomes. Subsequently, expansion of the centromeric domain and genome rearrangement gave rise to the holocentric condition. Later colonization of *Lusy*-type satellites allowed the maintenance of functional centromeres. M monocentric, H holocentric. Divergence time was obtained from the Timetree of Life (https://timetree.org/).

Furthermore, holocentric chromosomes can also originate from monocentromere spreading or chromosome rearrangements. The transition to holocentricity in *Luzula* could have been initiated earlier than the split of the *Luzula/Juncus* genus, since the repeat-based centromeres of *J. effusus* were recently described as an atypical monocentromere, with up to three centromere cores and different types of centromeric organization, resembling an almost widespread metapolycentric organization[23]. In *Luzula*, dramatic karyotype changes took place, resulting in reduced chromosome number. Although phylogenetic relationships are poorly resolved in the genus, descending dysploidy has been observed in species from different *Luzula* clades (*L. elegans* in the Marlenia clade and *L. purpureo-splendens* in the Nodulosae clade), indicating that this process of chromosomal evolution has occurred independently at least twice during the evolution of the genus[26]. Our results support dysploidy as the main driver of karyotype evolution in holocentric organisms (Supplementary Fig. 11), since fission and fusion events have been observed in sedges, leading to dysploid karyotypes in the holocentric genera *Rhynchospora*[4,6] and *Carex*[5,44], as well in some holocentric butterflies[45,46]. However, the fusion of ancestral chromosomes that resemble chromosomes from the sister genus *Juncus* resulting in the dysploid *L. sylvatica* is intriguing, since it involves a simultaneous shift of centromere organization. The 21 putative *Juncus* ancestral-like chromosomes merged into six *L. sylvatica* chromosomes while undergoing additional chromosome rearrangements, genomic reshuffling, and repetitive DNA turnover in the past ~60 million years of divergence[47]. Unlike the repeat-mediated chromosome fusions observed in *Rhynchospora*[4], fusion sites in *L. sylvatica* lack genomic footprints, suggesting another mechanism of chromosome fusions during karyotype evolution in *Luzula* and/or a masking over time by additional chromosomal rearrangements and dynamic repeat turnover.

Based on our findings, which are supported by the few ancestral centromeric regions found in both *J. effusus* and *L. sylvatica*, we propose a multistep model of evolutionary transition to holocentricity in the genus *Luzula* (Fig. 7). At first, a hypothesized change of the genomic processes responsible for centromere maintenance enables centromere spreading without inactivation, as observed in *Juncus*. Next, stepwise chromosome fusions of *Juncus*-like chromosomes could generate a 'protopolycentric' chromosome, which could gradually transition to a more diffused holocentric organization with tens to hundreds of discrete CENH3 domains through genome rearrangements and CENH3 spreading or seeding[11]. Subsequent centromere maturation can culminate with satellites invading these loci by a combination of satellite DNA library diversification and concerted evolution, as observed for *Tyba* repeats[39]. From this point of view, the presence of satellite-free centromeric units, the uneven distribution of centromeric satellite repeats and the cluster-like distribution of CENH3 and outer kinetochore proteins in *L. sylvatica* can be interpreted as intermediate stages of ongoing holocentromere maturation. An alternative model, with the first transition to holocentricity followed by stepwise fusion of chromosomes, cannot be discarded[4]. Further research can provide insight into the molecular mechanisms' adaptation taking place during the holocentric transition and its triggers.

## Methods

### Plant material

For cytogenetic analyses, plants from natural populations of *L. sylvatica* were collected in Cologne, Germany, and further cultivated under controlled greenhouse conditions (16 h daylight, 26 °C, >70% humidity). The ornamental plant *L. nivea* was commercially obtained (Dingers Gartencenter) and cultivated under controlled greenhouse conditions (16 h daylight, 20 °C).

### Genome assembly and Hi-C scaffolding

HiFi and Hi-C reads obtained through the Darwin Tree of Life database (www.darwintreeoflife.org)[28] were assembled using Hifiasm[48], available at https://github.com/chhylp123/hifiasm, following the command:

"*hifiasm -o output.asm -t 40 reads.fq.gz*". Preliminary assemblies were evaluated for contiguity and completeness with BUSCO[49] and QUAST[50].

Hi-C reads were first mapped to the primary contigs file obtained from the Hifiasm assembler using BWA[51] following the hic-pipeline (https://github.com/esrice/hic-pipeline). Hi-C scaffolding was performed using SALSA2 (https://github.com/marbl/SALSA)[52] with default parameters using '*GATC, GAATC, GATTC, GAGTC, GACTC*' as restriction sites. After testing several minimum mapping quality values of bam alignments, the final scaffolding was performed with MAPQ10. Following the automated scaffolding by SALSA2, several rounds of visual assembly correction guided by Hi-C heatmaps were performed. When regions showed multiple contact patterns, manual re-organization of the scaffolds was performed with Juicebox[53] and 3D-DNA assembly pipeline[54] to correct position/orientation and to obtain the six pseudomolecules.

Genome size estimate was obtained from HiFi reads using findGSE[55]. First, a histogram of k-mers was created using jellyfish[56], and then the findGSE R package was used for model fitting according to package documentation (https://github.com/schneebergerlab/findGSE).

## Chromatin immunoprecipitation (ChIP-seq) sequencing and analysis

ChIP experiments were performed following Hofstatter et al.[4]. In brief, *L. sylvatica* leaves were harvested and frozen in liquid nitrogen until sufficient material was obtained. The samples were fixed in 4% formaldehyde for 30 min and the chromatin was sonicated to enrich for 300 bp fragments. Then, 40 ng of sonicated chromatin was incubated with 2 ng of antibody overnight. Immunoprecipitation experiments were carried out for the rabbit anti-*L. elegans* CENH3 (LeCENH3)[57], rabbit anti-H3K4me3 (abcam, ab8580), and mouse anti-H3K9me2 (abcam, ab1220). Anti-LeCENH3 that was originally developed against 3-RTKHFSNRKSIPPKKQTPAK-23 peptide from *Luzula elegans* bears 65% similarity to the corresponding sequence 3-RTKHFSLRSRHPKKQRTAA-22 from *Luzula sylvatica* CENH3 (Gen-Bank: KJ934236.1). Recombinant rabbit IgG (abcam, ab172730) and no-antibody inputs were used as controls. Two experimental replications were also maintained for all the combinations. ChIP DNA was quality-controlled using the NGS-assay on a FEMTO-pulse (Agilent); next, an Illumina-compatible library was prepared for all immunoprecipitants with the Ovation Ultralow V2 DNA-Seq library preparation kit (Tecan Genomics) and single-end 1 ×150-bp reads were sequenced on a NextSeq 2000 (Illumina) device. For each library, an average of 20 million reads was obtained.

The raw sequencing reads were trimmed by Cutadapt[58] to remove low-quality nucleotides (with quality score less than 20) and adapters. Trimmed ChIPed 150-bp single-end reads were mapped to the respective reference genome with bowtie2[59], where all read duplicates were removed and only the single best-matching read was kept on the final alignment BAM file. ChIP vs input signal was calculated as the log2 ratio of read coverages normalized by reads per kilobase per million mapped reads (RPKM) using the bamCompare tool from deepTools package[60]. Averaged signal from both replicates was visualized using pyGenomeTracks[61].

Metaplots obtained by the plotProfile function from deepTools were used to compare the distribution with other genomic features[60]. To concretize enriched domains, we performed peak-calling by MACS3[62] and epic2[63] and filtered only the peaks identified by both tools in both replicates. This high stringency peak filtering approach was chosen to reduce the risk of including false positive CENH3 domains in subsequent analyses. Based on analysis of CENH3 peak clustering, peaks closer than 150 kb were merged to obtain uninterrupted centromeric units (code available on GitHub at https://github.com/437364/Repeat-based-holocentromeres-of-Luzula-sylvatica).

## Methylation sequencing and analysis

To analyze DNA methylation level, a sequencing library was prepared using NEBNext® Enzymatic Methyl-seq Kit (NEB; catalog number E7120S). Library sequencing was performed using NextSeq 2000 (Illumina) platform, obtaining ~20 M paired-end reads. Sequencing data was analyzed using the Bismarck pipeline[64] according to the toolkit documentation (https://felixkrueger.github.io/Bismark/bismark/). Coverage files for CpG, CHG, and CHH methylation contexts were converted to bigwig.

## Repeat characterization

Available Illumina reads from ENA was filtered by quality with 95% of bases equal to or above the quality cut-off value of 10 using RepeatExplorer2 pipeline (https://repeatexplorer-elixir.cerit-sc.cz/)[65]. The clustering was performed using the default settings of 90% similarity over 55% of the read length. For the comparative analyses, we performed an all-to-all similarity comparison across all species following the same approach. Because the genome size is unknown for some analyzed species, each set of reads was down-sampled to 1,000,000 for each species. Additionally, a subsample of eight species with known genome size were analyzed to compare the results. Samples from each species were identified with the four-letter prefixes shown in Table 1, and concatenated to produce datasets as input for RepeatExplorer2 graph-based clustering.

The automatic annotation of repeat clusters obtained by RepeatExplorer2 was manually inspected and reviewed, and was followed by recalculation of the genomic proportion of each repeat type when appropriate. DANTE and DANTE-LTR retrotransposon identification (Galaxy Version 3.5.1.1) pipeline was used to identify full-length LTR retrotransposons in the assembled genome, using a set of protein domains from REXdb[66]. All complete LTR-RTs contain GAG, PROT, RT, RH and INT domains, including some lineages encoding additional domains, such as chromodomains (CHD and CHDCR) from chromoviruses[67] or ancestral RNase H (aRH) from Tat elements[68]. DANTE_LTR retrotransposon filtering (Galaxy Version 3.5.1.1) was used to search for good quality retrotransposons, those with no cross-similarity between distinct lineages. This tool produced a GFF3 output file with detailed annotations of the LTR-RTs identified in the genome and a summary table with the numbers of the identified elements[66]. Overall repeat composition was calculated, excluding clusters of organelle DNA (chloroplast and mitochondrial DNA). Tandem sequences were identified using TAREAN[69]. All putative tandem sequences were compared for homology with DOTTER[70]. All tandem sequences were individually mapped to the genome by BLAST with 95% similarity using Geneious[71]. The mapped sequence files were converted to BED and used as an input track for a genome-wide overview with ShinyCircos using a 300 kb window[72]. Interstitial telomere sequences (ITS) were annotated on two stringency levels by searching for regions longer than 200 bp with at least 75 or 90% similarity to consensus *Arabidopsis*-type telomere arrays (monomer TTTAGGG) using Geneious[71], arrays closer than 10 kbp were merged.

## Characterization of centromeric units

Centromeric units from ChIP-seq analysis were grouped by chromosome and their size, count, and density were calculated. Next, centromeric units were overlapped with locations of satellites to obtain locations and extract sequences of functional array fragments (precise regions where centromeric satellites *Lusy1/Lusy2* and CENH3 domains overlap), nonfunctional arrays (whole *Lusy1/Lusy2* arrays not overlapping CENH3 domains), and satellite-free units (CENH3 domains not overlapping any satellites). Sequences of discrete arrays of each type were concatenated and their homogeneity assessed using ModDotPlot (https://github.com/marbl/ModDotPlot)[73]. Dyad symmetries were identified in each array using the EMBOSS palindrome tool[15] with

*nummismatches parameter set to 0.* Statistical significance of the increase of dyad symmetry abundance for functional arrays was tested using one-tailed Mann-Whitney U test from scipy package[74]. Proportions of functional and nonfunctional arrays as well as other genetic and epigenetic features in 100 kb windows were correlated using Spearman's rank correlation from scipy package and resulting correlation coefficients were plotted in a heatmap (code available on GitHub at https://github.com/437364/Repeat-based-holocentromeres-of-Luzula-sylvatica). Additional packages were used for data handling and visualization[75–79].

### Synteny analysis

The synteny analysis between *L. sylvatica* and *J. effusus* (as well as additional synteny of contig-level *L. sylvatica* assembly, see Supplementary Fig. 13) was performed with CoGe SynMap platform (https://genomevolution.org/coge/SynMap.pl)[80] and SyMAP v. 5.0.6[81]. For this analysis, CDS sequences, centromeric and telomeric repeats of both species were used. Synteny plots were obtained with GENESPACE[82]. Orthologs were identified following the steps: (1) using the BlastZ tool; (2) synteny analysis was performed using DAGChainer, using 25 genes as the maximum distance between two matches (-D) and 20 genes as the minimum number of aligned pairs (-A); (3) Quota Align Merge was used to merge syntenic blocks, with 50 genes as the maximum distance between them; and (4) orthologous and paralogous blocks were differentiated according to the synonymous substitution rate (Ks) using CodeML (where 2 was the maximum value of log10), and represented with different colors in the dot plot (Supplementary Fig. 10).

For the characterization of the regions involved in fusions, we followed Hofstatter et al.[4]. The synteny alignment between *L. sylvatica* and *J. effusus* genomes obtained in SyMAP allowed us to pin the putative regions around the borders of the fusion events. In order, to identify the underlying sequences at the fusion regions, we loaded annotation features for genes, TEs, and tandem repeats on SyMAP alignments. This allowed us to detect the sequence types in the putative fused regions. Further inspection and characterization of such regions were done by checking the genome coordinates and annotation features with Geneious[71].

To estimate the position of ancestral centromeres in *L. sylvatica* genome based on synteny with *J. effusus* (Supplementary Fig. 12e), we projected the position of the closest synteny blocks on both sides of the *J. effusus* centromere onto the *L. sylvatica* genome using the "2D" visualization of syntenic blocks in SyMAP. The region between these two projected coordinates was designated as a possible location of the ancestral centromere. However, due to numerous chromosomal rearrangements, the position of some ancestral centromeres could not be projected precisely, resulting in dramatically larger projected regions than the size of the original centromere (chromosome 6). Projected ancestral centromere regions were visualized using RIdeogram R package[83].

To verify that the chromosomal rearrangements discovered by synteny analysis are not a result of technical errors during the genome assembly scaffolding stage, we generated a synteny plot between *J. effusus* and individual large contigs of *L. sylvatica* (> 1 Mb) using GENESPACE[82].

To further analyze colocalization of genomic features, epigenetic marks, and fusion regions; we selected 50 kb regions upstream and downstream of syntenic block edges facing the fusion regions and also fusion regions where the space between two syntenic blocks was larger than 100 kb. To analyze whether these regions are enriched or depleted of specific features, 1000 rounds of random region distribution or random region distribution excluding satellite array locations were simulated (this was done to improve the reliability of the null distribution for features that are defined as not overlapping with satellite arrays, e.g. satellite-free CENH3 domains and genes)[84]

using *bedtools shuffle*. Then, overlap with all other studied features was calculated for simulated and real regions as a proportion of overlapping bases to all bases covered by the feature. The percentage of real overlap proportion in the distribution of simulated values was reported (code available on GitHub at https://github.com/437364/Repeat-based-holocentromeres-of-Luzula-sylvatica).

### Cytogenetic and immunostaining of CENH3 protein

Mitotic preparations were made from root meristems fixed in 4% paraformaldehyde and Tris buffer (10 mM Tris, 10 mM EDTA, 100 mM NaCl, 0.1% Triton, pH 7.5) for 30 min on ice in vacuum and for another 20 min only on ice. After washing twice in 1 x PBS for 10 min, the roots were digested in a cellulase-pectinase (2% w/v /20% v/v solution) containing PBS buffer and squashed in PBS. The coverslips were removed in liquid nitrogen and the slides were air-dried and stained in 2 μg/mL DAPI/Vectashield mounting medium for slide selection under the epifluorescence microscope. The slides with the highest number of cells in division were incubated in 3% (w/v) bovine serum albumin (BSA) containing 0.1% Triton X-100 in PBS. Immunostaining was performed using the primary antibodies rabbit anti-LeCENH3 (dilution 1:100)[57], rabbit anti-KNL1 (dilution 1:1000, GenScript, NJ, USA), rabbit anti-NDC80 (dilution 1:1000, Biomatik, ON, Canada)[29] and mouse anti-α-tubulin (dilution 1:100, Sigma-Aldrich, St. Louis, MO; catalog number T6199). Antibodies against KNL1 and NDC80 were originally developed using respective peptides EDHFFGPVSPSFIRPGRLSDC and EQGINARDAERMKRELQALEG from *Cuscuta* sp. These epitopes have a respective 55.6% and 57.1% similarity to peptides DDNFFGPVSAKFLKSGRFSDT and EQEVNLRD VDRMKREMQLIER identified by tblastn similarity search of *Cuscuta europaea* KNL1 and NDC80 protein sequences in *L. sylvatica* genome. As the secondary antibody, goat anti-Rabbit IgG antibody conjugated with Alexa Fluor 488 (Invitrogen; catalog number A27034), goat anti-rabbit conjugated with Rhodamine Red X (Jackson ImmunoResearch, catalog number: 111-295-144) or goat anti-mouse conjugated with Alexa Fluor 488 ( Jackson ImmunoResearch; catalog number 115-545-166) were used in a 1:500 dilution. Slides were incubated overnight at 4 °C, washed 3 times in 1×PBS and then the secondary antibody was applied, incubated at room temperature for 3 h and washed 3 times in 1×PBS. The slides were counterstained with 2 μg/mL DAPI/Vectashield mounting medium. Microscopic images were recorded using a Zeiss Axiovert 200 M microscope equipped with a Zeiss AxioCam CCD. Images of at least 5 cells were analyzed using the ZEN software (Carl Zeiss GmbH). Immuno-FISH was performed following Dias et al.[23], the immunostained slides were washed with 1xPBS for 15 min, postfixed in 4% paraformaldehyde in 1xPBS for 5 min, and then probed with the satellite *Lusy1* for 24 hours at 37 °C. Stringent washes were performed with 2× and 0.1× SSC at 42 °C to give a final stringency of ~76%.

Oligo probes from the most abundant tandem repeats *Lusy1* (GATCTCAAGAACACGTTATTTAGACTCGTCAAAGCA) and *Lusy2* (AAT TAATGACTAACACGATGCGAATTTCAATTTTTT) and the *Arabidopsis* telomeric sequence (TTTAGGG) were used for fluorescent in situ hybridization (FISH). Mitotic chromosomes from roots pretreated with 2 mM 8-hydroxyquinoline for 24 h at 4 °C and fixed with ethanol:acetic acid (3:1 v/v) for 2 h, were prepared using the air-drying method[38]. FISH was performed in denatured chromosomes at 75 °C for 5 min. The hybridization mixture contained formamide 50% (v/v), dextran sulphate 10% (w/v), 2 × SSC, and 50 ng/μL of each labelled probe. The slides were hybridized with this mixture for at least 24 hours at 37 °C[23]. Stringent washes were performed with 2× and 0.1× SSC at 42 °C to give a final stringency of ~76%. The slides were counterstained with 2 μg/mL DAPI in Vectashield (Vector) mounting medium. The images of at least 10 cells were captured as described above.

To analyze the chromatin ultrastructure, we applied super-resolution spatial structured illumination microscopy (3D-SIM) using

a 63x/1.40 Oil Plan-Apochromat objective of an Elyra PS.1 microscope system and the ZENBlack software (Carl Zeiss GmbH)[85]. Maximum intensity projections from image stacks were calculated from 3D-SIM image stacks. Zoom-in sections were presented as single slices to indicate the subnuclear chromatin structures at the super-resolution level.

## Reporting summary

Further information on research design is available in the Nature Portfolio Reporting Summary linked to this article.

## Data availability

The sequencing data generated in this study have been deposited in the NCBI database under the BioProject ID PRJNA1135980. The processed reference genomes, sequencing data, annotations and all tracks data generated in this study are available at Zenodo [https://zenodo.org/records/14007621]. The REXdb database Viridiplantae v.3.0 is publicly available at Github [https://github.com/repeatexplorer/rexdb]. Source data are provided with this paper.

## Code availability

The original code used in this study is available at GitHub [https://github.com/437364/Repeat-based-holocentromeres-of-Luzula-sylvatica][86].

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

## Acknowledgements

We thank Dra. Magdalena Vaio for her fruitful comments on the manuscript. We acknowledge the excellent technical assistance of Ursula Pfordt and Christina Philipp. This work was funded by the Max Planck Society (core funding to A.M.), by the Deutsche Forschungsgemeinschaft (DFG, grant no. MA 9363/3-1 to A.M.) and by the European Union (European Research Council Starting Grant, HoloRECOMB, grant no. 101114879 to A.M.). DFG founded this work under Germany's Excellence Strategy—EXC 2048/1–390686111 (A.M.). We thank the International Cooperation Program PROBRAL (CAPES/DAAD project number 88881.144086/2017-01) for the scholarship offered to Y.M.S. We thank the Darwin Tree of Life Project at the Wellcome Sanger Institute for making the data available (https://www.darwintreeoflife.org/project-resources). Computational resources for RepeatExplorer analysis were provided by the ELIXIR-CZ project (LM2023055), part of the international ELIXIR infrastructure. e-INFRA CZ project (ID:90254), supported by the Ministry of Education, Youth and Sports of the Czech Republic provided computational resources for the analysis of ChIP-seq data. The work of M.K. was supported by the Czech Science Foundation, grant no. GA24-11400S. E.K. was supported by the grant 21-00580S from the Czech Science Foundation. L.O. was supported by grant 2025440S from the Czech Science Foundation. A.P.H. and G.S. received productivity fellowship from CNPq (process numbers PQ-312852/2021-5 and PQ-312852/2021-5, respectively).

## Author contributions

Y.M.S.: Investigation, Validation, Formal analysis, Data Curation, Writing-Original draft preparation. M.K.: Validation, Formal analysis, Data Curation, Writing- Reviewing and Editing. L.O., P.N. and J.M.: Investigation, Formal analysis, Resources, Writing- Reviewing and Editing. B.H., V.S. and A.H.: Resources, Writing- Reviewing and Editing. E.K, A.P.H. and G.S.: Supervision, Resources, Writing- Reviewing and Editing. A.M.: Conceptualization, Supervision, Resources, Funding acquisition, Writing- Reviewing and Editing.

## Funding

## Competing interests

The authors declare no competing interests.
