## [Peer Review file · Nature Communications]

Repeat-based holocentromeres of the woodrush *Luzula sylvatica* reveal insights into the evolutionary transition to holocentricity

Corresponding Author: Dr André Marques

Version 0:

Reviewer comments:

Reviewer #1

(Remarks to the Author)

Mata-Sucre et al provide a comprehensive description of the genome and epigenome of *Luzula sylvatica* with emphasis on centromere structure. The data confirm the species is holocentric. This species, however is quite different from *Rhynchospora*, the genus they published on a couple years ago. Where *Rhynchospora* appears to be highly evolved with precisely sized and distributed centromere units where Tyba elements are closely tied to CENH3, *Luzula* is more jumbled, with different sized units and different underlying sequences. They describe two major centromere repeats and conclude that one of them (*Lusy1*) is specialized for functional centromere regions. By comparison to *Rhynchospora*, they suggest that *Luzula* is in an intermediate stage of evolution towards holocentricity. This work, like the early work from this group, is presented with high quality writing and excellent graphs and figures. I don't see any major problems here. It will be a fine addition to the centromere literature.

Minor comments

-Abstract. Second to last line should read *Juncus*-like chromosomes (not centromeres). Last line should read colonization of CENH3 regions by satellite DNA (not the reverse).

-Line 90. Be careful not to imply that HJURP is conserved.

-Figure 1e. (and Figure 6). People will wonder if the particularly large CENH3 areas are remnants of the monocentric regions from a monocentric ancestor. To address this question, arrows could be placed on the chromosome showing where the predicted centromeres from the *Juncus*-like relative would be located.

-Line 285. It would be useful to comment on why there was no NDC80 staining in *L. sylvatica*. I presume the gene is there. If it is just because the antibody does not recognize the protein in this species, can say that.

-Figure 4. The microscopy is excellent in this manuscript but the spindle staining in panel a is weak. It is not apparent where the spindle contacts KNL1 staining. Arrows would help. Similarly, in b, the projection makes it difficult to see where spindle is relative to KNL1. Single optical sections may be better here.

Figure 6. The authors argue in the next figure that the *Juncus* chromosomes fused and kept their centromeres to start the holocentric transition. If that is the case, we might expect the the locations of the *Juncus*-like centromere locations to be centromeric in *Luzula*. (if not, please explain). At the least, the position of the *Juncus* centromere should be noted, preferably with a line showing if the same region is centromeric in *Luzula*.

Reviewer #2

(Remarks to the Author)

In their manuscript titled "Repeat-based holocentromeres of the woodrush *Luzula sylvatica* reveal new insights into the

evolutionary transition from mono- to holocentricity”, the authors describe a genome assembly and characterization of the focal species, as well as a comparative analysis of genome synteny against a related species, *Juncus effusus*. I found the manuscript quite interesting, and in particular the genome assembly problem that arises with holocentromeres wherein a number of common heuristics used to assess assembly quality become less useful. That said, I'm not convinced that the authors have solved these issues in the present assembly for a number of reasons I describe below. At present I would recommend major revisions of the present manuscript so that these problems can be solved and the focal dynamics are characterized in a more convincing way.

1. I have now worked on a number of genomes both large and small, animal, plant, and fungi, where Hi-C scaffolding is applied to contigs (Illumina-, PacBio-, and Oxford Nanopore Technology (ONT)-based), and while I'm generally convinced that the method can effectively aggregate contigs into chromosomal groupings, I am not convinced that arranging the contigs into the correct order along the chromosome is particularly accurate. The authors do suggest some degree of manual curation but don't provide much detail about how this process was accomplished or what sort of corrections were made. These details need to be added.

2. Following upon point one, the authors need to either 1) provide a rationale for why they believe the ordering is correct, or 2) collect an additional source of data which will confirm this ordering. In the first case I've seen such analyses as gene distribution, though I suspect this wouldn't actually work given the nature of the holocentric centromeres. I think a much more useful and convincing type of evidence would be one or more of the following: 1) other forms of long-range sequencing (e.g., Bionano optical map, ONT ultra-long reads, etc.), 2) Linkage-map based ordering, or 3) population sequencing and LD based assessment of variants.

3. Without seeing the additions relating to points 1) and 2), I'm very skeptical about aspects of the synteny analysis. I am reasonably convinced that the authors can argue for major structural changes in the genome in the focal species relating to chromosomal fusions as compared to the outgroup, but the within chromosomal rearrangements could just as easily be the result of mis-scaffolding via Hi-C. Given so much of the interesting evolutionary dynamics related to holocentromeres is related to these sorts of rearrangements, having better certainty of the contig ordering is very important. Perhaps an initial step the authors could pursue is including a synteny analysis based on contigs alone.

Reviewer #3

(Remarks to the Author)

The manuscript by Mata-Sucre and colleagues reports a comprehensive characterization of the genome and the epigenome of the woodrush *Luzula sylvatica*, focusing on the organization of its holocentromeres. The authors assembled a high-quality chromosome-level genome assembly using publicly available PacBio HiFi reads and a Hi-C dataset. The quality of this genome sequence is demonstrated by its success in assembling satellite repeats. Two centromeric satellites, *Lusy1* and *Lusy2*, that are spread throughout all chromosomes were identified. Comparative analysis revealed that these satellite repeats are conserved among most *Luzula* species.

The authors then investigated the organization of centromeres in *L. sylvatica* by combining cytogenetic and molecular techniques. They proved that this species carries holocentromeres with CENH3 domains distributed across the entire length of all chromosomes. While most CENH3 domains were associated to satellite arrays, satellite-free CENH3 domains were also detected as well as non-centromeric satellite repeats. CENH3 domains typically arise in gene-desert regions which are associated with DNA methylation and heterochromatin markers.

The authors then investigated the synteny between the chromosomes of *L. sylvatica* and *Juncus effusus*, a rush species belonging to a different genus and carrying monocentric chromosomes. The genera *Juncus* and *Luzula* separated about 60 million years ago and belong to the family Juncaceae. While the karyotype of *J. effusus* resembles the putative ancestral karyotype of Juncaceae, the chromosomes of *L. sylvatica* were reshuffled by extensive karyotypic rearrangements and fusions. The authors propose a model suggesting that chromosomal fusions generated intermediate polycentric chromosomes which gave rise to holocentric chromosomes following spreading of satellite repeats and CENH3 domains.

Major critical points

- Lacking novelty is the main weakness of the manuscript. Although this is the first description of *L. sylvatica* centromeres and only a few holocentric species with centromeric repeats have been characterized so far, the same group already described a similar centromere organization in three beak-sedges belonging to the genus *Rhynchospora* (Hofstatter et al. Cell 2022). In their previous work, the authors compared genome and centromere organization between these holocentric sedges and the monocentric *J. effusus*, proposing that the presence of holocentric chromosomes may have facilitated chromosome fusions during karyotype evolution.

- Lines 133-149. The number of gaps in the assembly should be reported. The procedure used to annotate the genome and to produce the coverage tracks shown in Figure 1e is not well described. These details should be explained in the Methods section.

Lines 191-196. According to Figure 2a, both *Lusy1* and *Lusy2* are spread throughout all chromosomes being *Lusy1* more represented. Although the authors state that the distribution of *Lusy1* and *Lusy2* FISH signals is different no clear difference can be observed by looking at Figure 2b. The description of FISH images should be improved. In addition, DAPI staining should be shown.

Lines 277- 285. The description of the images shown in Figures 4 and S7 is not convincing. Maximum projection images are not sufficiently informative. Images from some single focal planes should be included to better observe the organization of KNL1, tubulin and *Lusy1* regions. Movies should be included as supplementary materials. For each experiment, it would be

better to include images showing complete metaphase spreads and zoomed-in images to highlight specific details. It is not clear why DAPI is only shown together with tubulin in Figure 4b. In Figures 4 and S7 different pseudo-colors are used for the same proteins. For clarity, it would be better to use the same colors in all images.

Double immunofluorescence experiments using anti-CENH3 and anti-KNL1 antibodies should be performed in *L. sylvatica* and *L. nivea* to verify the co-localization of these proteins. It would be interesting to perform a ChIP-seq with the anti-KNL1 antibody in *L. sylvatica* to test this colocalization at the molecular level.

- Lines 277-290. The lack of NDC80 signals in *L. sylvatica* should be investigated. It is not clear whether the lack of signals is due to technical problems, including antibody specificity, or to different kinetochore organizations in *L. sylvatica* and *L. nivea*. Additional genomic and proteomic analyses should be carried out to compare NDC80 in the two *Luzula* species.

Lines 340- 366. In Figures 6a and S10, syntenic blocks were identified between the chromosomes of *L. sylvatica* and *J. effusus*. However, these orthologies are not sufficient to support the model presented in Figure 7. Indeed, *L. sylvatica* chromosomes do not simply correspond to "fused" *J. effusus* chromosomes but complex rearrangements are also present. This is likely due to the fact that these species are not closely related but diverged more than 60 million years ago. This point is not well addressed and represents a limitation of the study and of the model. In addition, the authors did not identify specific sequence features at fusion regions. Given the evolutionary distance between the two species and the extension of the fusion regions analyzed, these conclusions are not surprising.

- In Figure 7 a model where fusions of monocentric chromosomes gave rise to intermediate polycentric chromosomes evolving into holocentrics is presented. The model and the Figure suffer from serious flaws. An alternative route, supported by the findings presented in previous work from the same group (Hofstatter et al 2022), implies that the transition to holocentricity precedes and favors chromosomal rearrangements and fusions.

As mentioned above, this model is based on the comparison between two species belonging to different genera that diverged a long time ago, about 60 million years. To test the hypothesis that fusions and holocentricity are related it would be necessary to compare *Luzula* species displaying high chromosomal number (for example *L. sudetica* and *L. pilosa*) with species with low chromosome number. In the Figure, sketches of nucleosome organizations are misleading. It is well known that, in monocentric centromeric domains, CENH3 nucleosomes are intermingled with H3 nucleosomes. The presence of two or three CENH3 domains in so-called Type 2 centromeres is possibly due to different CENH3 domains in the two homologous chromosomes and/or to sequence mis-assemblies in the underlying reference genome, as previously observed in some mammalian species. Therefore, the classification of Type 1 and Type 2 centromeres would require further investigation.

In the phylogenetic tree shown in the Figure, the hypothetical intermediate is placed in a sister clade of *Luzula*. However, according to the model, this intermediate should correspond to an ancestor of *Luzula* species. It is not clear how the dating of nodes was obtained.

- Lines 357-360. Only three interstitial telomeric loci in *L. sylvatica* genome were identified which do not colocalize with fusion regions. ITSs were searched with at least 90% similarity to TTTAGGG repetitions. Since fusions occurred several millions of years ago, it would be surprising to identify highly conserved repetitions at such loci. Less stringent parameters should be used to identify ITS loci possibly present at fusion regions.

Minor comments

- When abbreviations are used for the first time, the corresponding full name should be reported.
- When a species name is used for the first time, the entire name should be reported.
- Some information on KNL1, NDC80 and Cuscuta should be included in the Introduction.
- Line 153: "Online Methods" should probably be substituted with "Methods".
- Line 270: Figure S6b, not Figure 6e, should be mentioned here.
- Lines 358. The coordinates of ITS loci should be indicated.
- Lines 340- 366. Since the size of the identified fusion regions ranges from 10 kb to 8 Mb, they should be mentioned as "fusion regions" rather than "fusion points"
- Table S6: The content of the table should be clarified.

Reviewer #4

(Remarks to the Author)

Version 1:

Reviewer comments:

Reviewer #1

(Remarks to the Author)

The authors have addressed my major concerns, particularly regarding the ancient homologies of centromere regions. The authors have added new images that are said to show colocalization of KNL1 and tubulin. Unfortunately, I still do not see

evidence that KNL1 consistently interacts with tubulin (I could not view the videos with quicktime on my mac). The colocalization should be obvious. Either the authors should highlight specific regions where they think KNL1 contacts tubulin (with arrows), or say what appears to be true, that most KNL1 does not interact with tubulin at this stage of prophase. It is possible that the kinetochores do not fully engage spindles until metaphase.

Reviewer #2

(Remarks to the Author)

I appreciate the authors attention to my earlier concerns. Ultimately, I would not have found the applied revisions sufficient for acceptance had they not included the synteny analysis of an independent assembly of the focal species' genome. I fear that as more and more researchers are able to generate chromosomal level assemblies with long-read data alone, we will find that Hi-C scaffolding has led to numerous contig ordering errors. Luckily, in the present case, such errors appear to be absent. As such, I believe the present draft is satisfactory for publication.

Reviewer #3

(Remarks to the Author)

The answer to my comment on the lack of novelty of this manuscript is not satisfactory. Obviously, some novel findings are described but they do not represent a breakthrough as some of the ones previously published in top journals by the same group. In my opinion, having published on top journals should not guarantee future publications on top journals. I keep thinking that a more specialized journal would have been more appropriate for this manuscript. However, I must take note of the opinion of the other two reviewers and of the fact that the paper was sent out to referees.

Most other comments and changes inserted in the new version of the manuscript are appropriate.

The only point requiring further clarification is the description of the new Supplementary Figure S11. The Figure legend is wrong. Specific description of panels a, b, c, d and e should be added. In particular an explanation of panel e should be better detailed. Are entire chromosomes or portions of chromosomes depicted? What is the scale? Chromosome numbers for *J. effusus* are missing. The description of this Figure in the text should also be clarified.

Reviewer #4

(Remarks to the Author)

Version 2:

Reviewer comments:

Reviewer #1

(Remarks to the Author)

I am satisfied with the revisions relative to KNL1 and tubulin staining and have no further concerns.

RESPONSE LETTER

REVIEWER COMMENTS

Reviewer #1 (Remarks to the Author):

Mata-Sucre et al provide a comprehensive description of the genome and epigenome of *Luzula sylvatica* with emphasis on centromere structure. The data confirm the species is holocentric. This species, however is quite different from *Rhynchospora*, the genus they published on a couple years ago. Where *Rhynchospora* appears to be highly evolved with precisely sized and distributed centromere units where Tyba elements are closely tied to CENH3, *Luzula* is more jumbled, with different sized units and different underlying sequences. They describe two major centromere repeats and conclude that one of them (Lusy1) is specialized for functional centromere regions. By comparison to *Rhynchospora*, they suggest that *Luzula* is in an intermediate stage of evolution towards holocentricity. This work, like the early work from this group, is presented with high quality writing and excellent graphs and figures. I don't see any major problems here. It will be a fine addition to the centromere literature.

ANSWER:

Minor comments

-Abstract. Second to last line should read *Juncus*-like chromosomes (not centromeres). Last line should read colonization of CENH3 regions by satellite DNA (not the reverse).

ANSWER: We agree with this comment, and we corrected the abstract. Now is possible to read “We propose that the transition to holocentricity in *Luzula* involves: (i) fusion of small chromosomes resembling *Juncus*-like chromosomes; (ii) expansion of atypical centromeric units; and (iii) colonization of CENH3-interacting satellite DNA for centromere stabilization.”

-Line 90. Be careful not to imply that HJURP is conserved.

ANSWER: Thanks for point this out, we clarify that the specific HJURP protein is well characterized in humans.

-Figure 1e. (and Figure 6). People will wonder if the particularly large CENH3 areas are remnants of the monocentric regions from a monocentric ancestor. To address this question, arrows could be placed on the chromosome showing where the predicted centromeres from the *Juncus*-like relative would be located.

ANSWER: We have estimated the locations of ancestral centromeres in *Luzula sylvatica* by projecting the coordinates of syntenic blocks bordering centromeres in *J. effusus* onto *L. sylvatica* chromosomes, now as new Supplementary Figure S11. The resulting projection

shows that at least four centromere units appear to be conserved between *J. effusus* and *L. sylvatica*. However, due to numerous chromosomal rearrangements, the position of some ancestral centromeres could not be projected precisely, resulting in dramatically larger projected regions than the size of the original centromere (chromosome 6). We have included these results in the text on page 17 lines 409-415.

New Figure S11: Conservation of the ancestral centromere position between *Luzula sylvatica* and *Juncus effusus*. (a) Fine-scale synteny analysis revealed that at least two large

centromere units, particularly on chromosome 1, one on chromosome 3 and one on chromosome 4 appear to be conserved between *J. effusus* and *L. sylvatica* genomes. **(b)** Projection of the coordinates of syntenic blocks bordering centromeres in *J. effusus* were plotted in *L. sylvatica* genome. Conserved centromeres between *J. effusus* and *L. sylvatica* are highlighted with asterisks.

-Line 285. It would be useful to comment on why there was no NDC80 staining in *L. sylvatica*. I presume the gene is there. If it is just because the antibody does not recognize the protein in this species, can say that.

ANSWER: Indeed, our analysis of kinetochore protein annotation by blast and subsequent search using Mercator software confirmed the presence of the NDC80 gene in the genome of *L. sylvatica* (data not shown). In this case, the absence of signals could be attributed to other reasons such as: the low similarity of sequences between the immunodetected target NDC80 protein, which differs from that of *Cuscuta* (for which the antibody was developed) by 57%. It is also described that the detection of NDC80 is more difficult compared to KNL1, which is probably due to the high proportion of amino acid residues that are modified by formaldehyde during the fixative treatment (Oliveira et al. 2024). We have included this clarification in Page 14 line 327-329. It now reads: “Absence of NDC80 immunosignals in *L. sylvatica* could be due either to low amino acid sequence similarity with the target sequence developed in *Cuscuta*, or due to sensitivity of the protein during the cell fixation process, as discussed by Oliveira *et al.* (2024).”

-Figure 4. The microscopy is excellent in this manuscript but the spindle staining in panel a is weak. It is not apparent where the spindle contacts KNL1 staining. Arrows would help. Similarly, in b, the projection makes it difficult to see where spindle is relative to KNL1. Single optical sections may be better here.

ANSWER: As recommended, we have included additional inserts in the image that allow a better observation of microtubule attachment sites in KNL1. In addition, supplementary videos were included to better visualize the 3D display of signal in the cell.

Figure 6. The authors argue in the next figure that the *Juncus* chromosomes fused and kept their centromeres to start the holocentric transition. If that is the case, we might expect the the locations of the *Juncus*-like centromere locations to be centromeric in *Luzula*. (if not, please explain). At the least, the position of the *juncus* centromere should be noted, preferably with a line showing if the same region is centromeric in *Luzula*.

ANSWER: As shown in the new Figure S11 above, the projecting *J. effusus* centromeres onto *L. sylvatica* genome, demonstrate few cases where the centromere location appears conserved. In other cases, the projected location of ancestral centromere does not correspond to contemporary large centromeric units. This can be explained by the proposed dynamic nature

of the centromere in *Luzula*, where the ancestral centromeres undergo reorganization into smaller, evenly spaced units over time. Furthermore, projection cannot be reliably established for all *J. effusus* centromeres due to numerous chromosomal rearrangements, making the conclusions derived from these projections more speculative.

Reviewer #2 (Remarks to the Author):

In their manuscript titled “Repeat-based holocentromeres of the woodrush *Luzula sylvatica* reveal new insights into the evolutionary transition from mono- to holocentricity”, the authors describe a genome assembly and characterization of the focal species, as well as a comparative analysis of genome synteny against a related species, *Juncus effusus*. I found the manuscript quite interesting, and in particular the genome assembly problem that arises with holocentromeres wherein a number of common heuristics used to assess assembly quality become less useful. That said, I’m not convinced that the authors have solved these issues in the present assembly for a number of reasons I describe below. At present I would recommend major revisions of the present manuscript so that these problems can be solved and the focal dynamics are characterized in a more convincing way.

1. I have now worked on a number of genomes both large and small, animal, plant, and fungi, where Hi-C scaffolding is applied to contigs (Illumina-, PacBio-, and Oxford Nanopore Technology (ONT)-based), and while I’m generally convinced that the method can effectively aggregate contigs into chromosomal groupings, I am not convinced that arranging the contigs into the correct order along the chromosome is particularly accurate. The authors do suggest some degree of manual curation but don’t provide much detail about how this process was accomplished or what sort of corrections were made. These details need to be added.

ANSWER: We recognize that arranging contigs in the correct order along holocentric chromosomes can be challenging. However, we use Hi-C reads that are very accurate and provide detailed contact maps as a guide. Additionally, after automated scaffolding by SALSA, several rounds of visual assembly correction guided by Hi-C heatmaps were performed. When regions showed multiple contact patterns, manual re-organization of the scaffolds was performed with Juicebox and 3D-DNA assembly pipeline to correct position/orientation and to obtain the six pseudomolecules. This description was included in the Methods section, Page 24 lines 638-641.

2. Following upon point one, the authors need to either 1) provide a rationale for why they believe the ordering is correct, or 2) collect an additional source of data which will confirm this ordering. In the first case I’ve seen such analyses as gene distribution, though I suspect this wouldn’t actually work given the nature of the holocentric centromeres. I think a much more useful and convincing type of evidence would be one or more of the following: 1) other forms of long-range sequencing (e.g., Bionano optical map, ONT ultra-long reads, etc.), 2) Linkage-map based ordering, or 3) population sequencing and LD based assessment of variants.

ANSWER: We understand your concern and we agree that long-read sequencing improves the quality of the assemblies. However, combination from PacBio-HiFi and Hi-C sequencing generate long high-fidelity reads achieving an accuracy of more than 99.8%, turning this approach one of the most widely used sequencing methodologies nowadays. In addition, our cytogenetic mapping of the satellite DNA sequences showed a pattern of uneven enrichment along the chromosomes as observed *in silico*, providing additional evidence supporting the satellite array pattern obtained in genome assembly. Nevertheless, to ensure the accuracy of our ordering, we compared our assembly with the *Luzula sylvatica* genome assembly of similar size (444.5 megabases and 6 chromosomal pseudomolecules) recently published by the Darwin Tree of Life project (Goodwin et al. 2024) obtained using a different scaffolding method (YaHS). As shown in the figure below, this comparison demonstrates the reproducibility of our genome assembly method.

Review response figure 1: Genome synteny between our *L. sylvatica* HiC-scaffolded genome (middle) compared to the one released by the Darwin Tree of Life project (top), using different scaffolding methods, and with the *Juncus effusus* (bottom). Please note the very same order of arrangements in both *L. sylvatica* genome assemblies, supporting the accurate scaffolding of its chromosomes.

3. Without seeing the additions relating to points 1) and 2), I'm very skeptical about aspects of the synteny analysis. I am reasonably convinced that the authors can argue for major structural changes in the genome in the focal species relating to chromosomal fusions as compared to the outgroup, but the within chromosomal rearrangements could just as easily be the result of mis-scaffolding via Hi-C. Given so much of the interesting evolutionary dynamics related to holocentromeres is related to these sorts of rearrangements, having better certainty of the contig ordering is very important. Perhaps an initial step the authors could pursue is including a synteny analysis based on contigs alone.

ANSWER: We appreciate this suggestion for improving the support for our karyotype evolution model. We have now compared the synteny of large contigs from the *Luzula sylvatica* assembly (size > 1Mbp) with *Juncus effusus* assembly (where the large contigs by themselves are chromosome-level; see Hofstatter et al. 2022). As is demonstrated in the figure below, thanks to the high N50, contig synteny confirms several fusions and numerous small chromosomal rearrangements.

New Figure S12: Genome synteny between *L. sylvatica* and *J. effusus* contigs. Genome synteny patterns between individual large *L. sylvatica* contigs and *J. effusus* (chromosome-level) contigs shows that the observed fusions and genome rearrangements are not a result of erroneous scaffolding.

Reviewer #3 (Remarks to the Author):

The manuscript by Mata-Sucre and colleagues reports a comprehensive characterization of the genome and the epigenome of the woodrush *Luzula sylvatica*, focusing on the organization of its holocentromeres. The authors assembled a high-quality chromosome-level genome assembly using publicly available PacBio HiFi reads and a Hi-C dataset. The quality of this genome sequence is demonstrated by its success in assembling satellite repeats. Two centromeric satellites, *Lusy1* and *Lusy2*, that are spread throughout all chromosomes were identified. Comparative analysis revealed that these satellite repeats are conserved among most *Luzula* species.

The authors then investigated the organization of centromeres in *L. sylvatica* by combining cytogenetic and molecular techniques. They proved that this species carries holocentromeres with CENH3 domains distributed across the entire length of all chromosomes. While most CENH3 domains were associated to satellite arrays, satellite-free CENH3 domains were also detected as well as non-centromeric satellite repeats. CENH3 domains typically arise in gene-desert regions which are associated with DNA methylation and heterochromatin markers.

The authors then investigated the synteny between the chromosomes of *L. sylvatica* and *Juncus effusus*, a rush species belonging to a different genus and carrying monocentric chromosomes. The genera *Juncus* and *Luzula* separated about 60 million years ago and belong to the family Juncaceae. While the karyotype of *J. effusus* resembles the putative ancestral karyotype of Juncaceae, the chromosomes of *L. sylvatica* were reshuffled by extensive karyotypic rearrangements and fusions. The authors propose a model suggesting that chromosomal fusions generated intermediate polycentric chromosomes which gave rise to holocentric chromosomes following spreading of satellite repeats and CENH3 domains.

Major critical points

- Lacking novelty is the main weakness of the manuscript. Although this is the first description of *L. sylvatica* centromeres and only a few holocentric species with centromeric repeats have been characterized so far, the same group already described a similar centromere organization in three beak-sedges belonging to the genus *Rhynchospora* (Hofstatter et al. Cell 2022). In their previous work, the authors compared genome and centromere organization between these holocentric sedges and the monocentric *J. effusus*, proposing that the presence of holocentric chromosomes may have facilitated chromosome fusions during karyotype evolution.

ANSWER: Our group has indeed previously contributed with groundbreaking results describing the evolution of holocentric genomes in high quality journals, and this manuscript is not an exception. This study not only brings the first description of holocentromeres in *Luzula sylvatica*, but also provides new insights into the repeat-based organization of their holocentromeres, which are different from the recently described holocentromeres of *Chionographis japonica* (Kuo et al. 2023) and the repeat-lacking holocentromere of *Myristica fragrans* (Kuo et al. 2024). These results support the hypothesis that the organization of (holo)centromeres is very diverse and the transition in each group may have occurred under its own evolutionary pathway. Additionally, we show that the centromeres of *Luzula* species, a young genus in Juncaceae, are enriched by two distinct satellite types and that (epi)genomic traits suggest their competition for the centromere colonization. In contrast to our previous

work on *Rhynchospora*, this work elucidates the genome evolutionary history between two sister genera with different centromere organizations, offering valuable insights into the transition and diversity of centromeres that share a close common ancestry.

We know that centromere organization is crucial for understanding reproductive mechanisms, and this is particularly true for holocentric species, whose evolution led to specific adaptations over time. Thus, our findings highlight the unique evolutionary pathways of species within *Luzula* genus, which do not follow the previously established centromere organization patterns.

- Lines 133-149. The number of gaps in the assembly should be reported. The procedure used to annotate the genome and to produce the coverage tracks shown in Figure 1e is not well described. These details should be explained in the Methods section.

ANSWER: The number of gaps were included in Table S1 142 Gaps totaling 2557 Ns. Regarding to the annotation, the DANTE_LTR retrotransposon (RT) identification tool (Galaxy Version 3.5.1.1) was used for the identification and annotation of the nucleotide sequences of complete LTR-RTs using the complete DANTE output. In this case all complete LTR-RTs contain GAG, PROT, RT, RH and INT domains, including some lineages encoding additional domains, such as chromodomains (CHD and CHDCR) from chromoviruses (Neumann et al. 2011) or ancestral RNase H (aRH) from Tat elements (Neumann et al. 2019). DANTE_LTR retrotransposon filtering (Galaxy Version 3.5.1.1) was used to search for good quality retrotransposons, those with no cross-similarity between distinct lineages. This tool produced a GFF3 output file with detailed annotations of the LTR-RTs identified in the genome and a summary table with the numbers of the identified elements. Tandem sequences were identified using TAREAN (Novák et al. 2017). All putative tandem sequences were compared for homology with DOTTER (Sonnhammer and Durbin 1995). All tandem sequences were individually mapped to the genome by BLAST with 95% similarity using Geneious (Kearse et al. 2012). The mapped sequence files were converted to BED and used as an input track for a genome-wide overview with ShinyCircos using a 100kb window (Yu et al. 2018). We include this more detailed description in the revised version of the manuscript, section Material and Methods Pages 27-28, lines 731-755.

Lines 191-196. According to Figure 2a, both *Lusy1* and *Lusy2* are spread throughout all chromosomes being *Lusy1* more represented. Although the authors state that the distribution of *Lusy1* and *Lusy2* FISH signals is different no clear difference can be observed by looking at Figure 2b. The description of FISH images should be improved. In addition, DAPI staining should be shown.

ANSWER: We have improved the description of the FISH figures in the text and included the DAPI image in **Fig. 2b**. We highlight in page 9 Lines 214-224 that *Lusy1* presents a linear distribution along most chromosomes.

It now reads: “Remarkably, we observed that *Lusy1* shows a line-like distribution across the entire length of each sister chromatid (**Fig. 2b**), in a similar pattern to other holocentromeric repeats (Marques *et al.* 2015). In contrast, despite the presence of *Lusy2* in all chromosomes, it differs from this linear pattern, and shows a more diffuse and dispersed pattern of signal

across chromosomes. *Lusy2* appears to be present in areas where *Lusy1* is less enriched, suggesting a complementary distribution between the two satellites (**Fig. 2b**). Furthermore, in interphase nuclei, *Lusy1* signals are more focused compared to more disperse *Lusy2* signals, with clear occurrences of co-localization as well as regions where *Lusy1* and *Lusy2* signals do not overlap (**Fig. 2b**, bottom). These results suggest that although *Lusy1* and *Lusy2* may occupy shared regions, they also maintain distinct territories within chromatin, further supporting the idea of their distinct roles in chromosomal organization.”

Lines 277- 285. The description of the images shown in Figures 4 and S7 is not convincing. Maximum projection images are not sufficiently informative. Images from some single focal planes should be included to better observe the organization of KNL1, tubulin and *Lusy1* regions. Movies should be included as supplementary materials. For each experiment, it would be better to include images showing complete metaphase spreads and zoomed-in images to highlight specific details. It is not clear why DAPI is only shown together with tubulin in Figure 4b. In Figures 4 and S7 different pseudo-colors are used for the same proteins. For clarity, it would be better to use the same colors in all images.

ANSWER: Thank you for your detailed comments on Figures 4 and S7. As suggested by reviewer 1 as well, we have included individual zoom plane images in addition to the full view images. This allows a more detailed observation of the organization of the KNL1 and tubulin regions in the chromosomes of *L. sylvatica*. We have also added movies as supplementary material to provide a dynamic view of these structures. We have also included markers highlighting the locations where KNL1 and *Lusy1* localization sites are highlighted.

Regarding DAPI staining, we have revised the figures to systematically include DAPI in all relevant sections and have standardized the pseudocolors used for the same proteins in all images in Figures 4 and S7.

Double immunofluorescence experiments using anti-CENH3 and anti-KNL1 antibodies should be performed in *L. sylvatica* and *L. nivea* to verify the co-localization of these proteins. It would be interesting to perform a ChIP-seq with the anti-KNL1 antibody in *L. sylvatica* to test this colocalization at the molecular level.

ANSWER: We agree that double immunostaining with anti-CENH3 and anti-KNL1 antibodies in *L. sylvatica* would be interesting to show in this manuscript, due to the localization of KNL1 with the *Lusy* satellite. Therefore, we have included these experiments in Figure 4, where the co-localization of these proteins is demonstrated. In addition, we have discussed these findings in the text to highlight their importance. Regarding the ChIP-seq experiment with the anti-KNL1 antibody, we agree that it would provide valuable molecular information on the molecular bases of these proteins. However, KNL1-ChIP-seq did not work so far in our hands, which might be due to its outer kinetochore nature to be successfully crosslinked to the nucleosomal DNA. We hope that this reviewer understands the limitation of this experiment and that the KNL1-ChIP-seq is not essential for this work.

- Lines 277-290. The lack of NDC80 signals in *L. sylvatica* should be investigated. It is not clear whether the lack of signals is due to technical problems, including antibody specificity, or to different kinetochore organizations in *L. sylvatica* and *L. nivea*. Additional genomic and proteomic analyses should be carried out to compare NDC80 in the two *Luzula* species.

ANSWER: As commented for the previous reviewer, our analysis of kinetochore protein annotation by blast and subsequent search using Mercator software confirmed the presence of this protein in the genome of *L. sylvatica*. So, the absence of signals could be attributed to other reasons such as: the low similarity of sequences between the immunodetected target NDC80 protein, which differs from that of *Cuscuta* (for which the antibody was developed) by 57%. It is also described that the detection of NDC80 is more difficult compared to KNL1, which is probably due to the high proportion of amino acid residues that are modified by formaldehyde during the fixative treatment (Oliveira et al. 2024). We have included this clarification in Page 14 line 327-329. It now reads: “Absence of NDC80 immunosignals in *L. sylvatica* could be due either to low amino acid sequence similarity with the target sequence developed in *Cuscuta*, or due to sensitivity of the protein during the cell fixation process, as discussed by Oliveira et al. (2024).”

Lines 340- 366. In Figures 6a and S10, syntenic blocks were identified between the chromosomes of *L. sylvatica* and *J. effusus*. However, these orthologies are not sufficient to support the model presented in Figure 7. Indeed, *L. sylvatica* chromosomes do not simply correspond to “fused” *J. effusus* chromosomes but complex rearrangements are also present. This is likely due to the fact that these species are not closely related but diverged more than 60 million years ago. This point is not well addressed and represents a limitation of the study and of the model. In addition, the authors did not identify specific sequence features at fusion regions. Given the evolutionary distance between the two species and the extension of the fusion regions analyzed, these conclusions are not surprising.

ANSWER: It is indeed apparent that the evolutionary distance between the two species obfuscates the signature of ancestral fusions. However, the chromosomal rearrangements disrupt the syntenic blocks to a varying degree across the *L. sylvatica* genome. On chromosome 1, we observe large uninterrupted syntenic blocks in combination with possibly conserved ancestral centromere position and overall irregular centromeric units (Fig. S4), while on the chromosome 6, syntenic blocks are broken up due to chromosomal rearrangements and centromeric units are more uniform in their size and spacing.

This is in accordance with our model, which necessitates that chromosomes in various stages of centromere reorganization are subject to fusions.

- In Figure 7 a model where fusions of monocentric chromosomes gave rise to intermediate polycentric chromosomes evolving into holocentrics is presented. The model and the Figure suffer from serious flaws. An alternative route, supported by the findings presented in previous work from the same group (Hofstatter et al 2022), implies that the transition to holocentricity precedes and favors chromosomal rearrangements and fusions.

As mentioned above, this model is based on the comparison between two species belonging to different genera that diverged a long time ago, about 60 million years. To test the hypothesis that fusions and holocentricity are related it would be necessary to compare *Luzula* species displaying high chromosomal number (for example *L. sudetica* and *L. pilosa*) with species with low chromosome number. In the Figure, sketches of nucleosome organizations are misleading. It is well known that, in monocentric centromeric domains, CENH3 nucleosomes are intermingled with H3 nucleosomes. The presence of two or three CENH3 domains in so-called Type 2 centromeres is possibly due to different CENH3 domains in the two homologous chromosomes and/or to sequence mis-assemblies in the underlying reference genome, as previously observed in some mammalian species. Therefore, the classification of Type 1 and Type 2 centromeres would require further investigation.

In the phylogenetic tree shown in the Figure, the hypothetical intermediate is placed in a sister clade of *Luzula*. However, according to the model, this intermediate should correspond to an ancestor of *Luzula* species. It is not clear how the dating of nodes was obtained.

ANSWER: Indeed, the hypothesis that holocentricity facilitates chromosomal rearrangements and fusions is supported by the findings of our previous work (Hofstatter et al. 2022) and other work in holocentric species (Marques-Corro et al. 2019; Escudero et al. 2023). These studies emphasize the role of holocentricity in promoting chromosomal stability during extensive rearrangements, which we agree could be a critical aspect of their evolutionary advantage. However, these studies could not search for centromere-type transition determinants since all sedges are holocentric.

We want to make clear that our model is just a hypothesis based on our results, which is further supported by the few ancestral centromeric regions found in both *J. effusus* and *L. sylvatica*. Despite their long-time divergence (>60 mya) finding centromeres in conserved regions in both genomes is remarkable. Since the high chromosome number found in *Luzula* species might be a result of polyploidy, we think that testing this hypothesis would be better by inducing chromosome fusion in *Juncus*. Given their very small chromosomes it is likely that occasional dicentric chromosomes could still be transiently stable and function as a protopolycentric. However, we hope that this reviewer agrees that this is beyond the scope of the present study and could be planned for future research.

We recognize that comparison between species of different genera that diverged approximately 60 million years ago may not fully capture the complexity of centromeric evolution, but so far, few such close sister genera with different centromeric architectures have been reported in plants, so the results of this study bring insight into their evolution.

Regarding the nucleosome organization in monocentric centromeres and the placement of the hypothetical intermediate in the phylogenetic tree depicted in the figure, we agree that the current representation might be oversimplified and misleading. We have revised the figure to better reflect the current understanding of centromere structure, ensuring a more accurate depiction of nucleosome organization. Having said that we have reformulate our model and also the last paragraph of our discussion to lower down our hypothesis. Furthermore, we also provide a sentence with an alternative model.

- Lines 357-360. Only three interstitial telomeric loci in *L. sylvatica* genome were identified which do not colocalize with fusion regions. ITSs were searched with at least 90% similarity to TTTAGGG repetitions. Since fusions occurred several millions of years ago, it would be surprising to identify highly conserved repetitions at such loci. Less stringent parameters should be used to identify ITS loci possibly present at fusion regions.

ANSWER: We have now included regions with >75% similarity to telomeric arrays. Only the continuous regions of telomeric sequences longer than 200 bp were retained, telomeric arrays with distance <100 kbp were merged. This has revealed additional interstitial telomere-like arrays. Possible deteriorated ITS were found in the regions bordering (within ~50 kb) both sides of ancestral *J. effusus*-like chromosome 21 (present on *L. sylvatica* chromosome 1) and distal end of ancestral chromosome 8 block on contemporary chromosome 5. In total, this approach yielded 36 regions, 11 of which are within 5 Mb from the contemporary chromosome end, 3 are within ~50 kb of a fusion point block end and 21 elsewhere in the genome. The presence of cases of seeming expansion of ITS sequences (such as the ancestral chromosome 17 on contemporary chromosome 2) suggests that this lower similarity threshold is no longer specific and identifies microsatellites unrelated to the telomere. These results were included in the revised version of the manuscript, the image was included in the supplementary files (Fig. S13).

New Figure S13: Interstitial telomeric sites observed in the genome of *L. sylvatica*.

Syntenic regions with the *Juncus effusus* genome are shown as bars on the chromosomes, colors correspond to individual chromosomes. Ends of contigs (assembly gaps) and annotated telomeric regions on the 90% and 75% similarity threshold are marked along the chromosomes. Possible deteriorated ITS were found in the regions bordering (within ~50 kb) both sides of ancestral *J. effusus*-like chromosome 21 (present on *L. sylvatica* chromosome 1) and distal end of ancestral chromosome 8 block on contemporary chromosome 5.

Minor comments

- When abbreviations are used for the first time, the corresponding full name should be reported.

ANSWER: We appreciate that you noticed this, all abbreviations were checked and, when first mentioned, were spelled out in full. This includes centromeric histone H3 variant (CENH3), chromatin conformation capture (Hi-C), Benchmarking Universal Single-Copy Orthologs (BUSCO), chromatin immunoprecipitation followed by sequencing (ChIP-seq), Domain based Annotation of Transposable Elements (DANTE), DANTE for Long terminal repeat (DANTE-LTR), Tandem Repeat Analyzer (TAREAN), Fluorescent in situ hybridization (FISH) and immunostaining followed by fluorescent in situ hybridization (Immuno-FISH). The abbreviation SatDNA corresponding to satellite DNA has been deleted to ensure the flow of the text.

- When a species name is used for the first time, the entire name should be reported.

ANSWER: Species names were corrected to their full names when first mentioned.

- Some information on KNL1, NDC80 and *Cuscuta* should be included in the Introduction.

ANSWER: We agree with the comment and a paragraph explaining the key results on the kinetochore proteins in *Cuscuta* was included in the introduction in Page 4 lines 73-80. It can now be read: “Numerous evolutionary models have been proposed to explain the emergence of holocentricity from monocentric ancestors, which include alterations/loss/emergence of kinetochore genes or centromeric repetitive sequences during the process (Drinnenberg *et al.* 2014; Senaratne *et al.* 2021; Neumann *et al.* 2023; Kuo *et al.* 2024). In the *Cuscuta* genus of the Convolvulaceae family, the transition to holocentricity was associated with not just massive changes in the kinetochore component localization on the chromosomes, but also with a loss/truncation/alteration of some important representatives of the KMN complex such as KNL2, KNL1, ZWINT1, MIS12 and NDC80 (Neumann *et al.*, 2023).”

- Line 153: “Online Methods” should probably be substituted with “Methods”.

ANSWER: we have corrected this typo.

- Line 270: Figure S6b, not Figure 6e, should be mentioned here.

ANSWER: Yes, we have corrected this figure call to Fig. S6e.

- Lines 358. The coordinates of ITS loci should be indicated.

ANSWER: A new supplementary figure, Fig. S13, includes all the positions of the interstitial telomeric regions and the syntenic regions of the *J. effusus* genome. Additionally, a bed file with coordinates was included in the public repository associated with the publication.

- Lines 340- 366. Since the size of the identified fusion regions ranges from 10 kb to 8 Mb, they should be mentioned as “fusion regions” rather than “fusion points”

ANSWER: Indeed, the size of the identified fusion ‘points’ are regions larger than 10 kb so it is more appropriate to refer to them as "fusion regions". We have corrected this in the text.

- Table S6: The content of the table should be clarified.

ANSWER: Certainly, a more detailed explanation was included in the table caption. It now reads: ‘Comparative analysis of the shared repetitive sequences between *Luzula* species genomes. The different lineages of elements were grouped by cluster in a hierarchical manner following the Repeatexplorer default parameters. Proportions are expressed as percentages. Code names correspond to Larcu: *Luzula arcuata*, Lcamp: *Luzula campestris*, Leleg: *Luzula elegans*, Lluzu: *Luzula luzuloides*, Lmtsf: *Luzula multiflora subsp. frigida*, Lniva: *Luzula nivalis*, Lnvea: *Luzula nivea*, Lparv: *Luzula parviflora*, Lpilo: *Luzula pilosa*, Lspic: *Luzula spicata*, Lsude: *Luzula sudetica*, Lsylv: *Luzula sylvatica*, Lwahl: *Luzula wahlenbergii*.’

Reviewer #4 (Remarks to the Author):

ANSWER: Thank you for participating in the peer review process, the revisions and comments were addressed above one by one.

REVIEWER COMMENTS

Reviewer #1 (Remarks to the Author):

The authors have addressed my major concerns, particularly regarding the ancient homologies of centromere regions. The authors have added new images that are said to show colocalization of KNL1 and tubulin. Unfortunately, I still do not see evidence that KNL1 consistently interacts with tubulin (I could not view the videos with quicktime on my mac). The colocalization should be obvious. Either the authors should highlight specific regions where they think KNL1 contacts tubulin (with arrows), or say what appears to be true, that most KNL1 does not interact with tubulin at this stage of prophase. It is possible that the kinetochores do not fully engage spindles until metaphase.

Answer: We thank the reviewer for this suggestion. Arrows have been included in image 4b to indicate the regions where colocalization between KNL1 and tubulin is most evident. Furthermore, an overview of this interaction can be observed in Supplementary Movie 1, as illustrated in the newly added Suppl. Figure 7 is displayed below. In this image, we have indicated the locations where these interactions are observed. We have now converted the videos to mp4 format, which should now be possible to be played on QuickTime.

Supplementary Figure 7: Localization of KNL1 and tubulin in *Luzula sylvatica* mitotic chromosomes. The images in (a–d) represent 3D-SIM super-resolution differential plane sections of the Suppl. Movie 1. Please note the interaction between KNL1 and tubulin (indicated by arrows).

Moreover, our observations indicate that even in cells exhibiting incomplete metaphase chromosome condensation, as evidenced in the video, microtubules persist in binding to KNL1. We believe that the highlighted regions in the updated images provide more compelling evidence for this interaction. While the interaction may be weak in some cases, the evidence nevertheless indicates that microtubules actively bind to KNL1, suggesting a potential role for these proteins in facilitating early kinetochore-microtubule interactions. This is a topic that would benefit from further investigation in future research.

Reviewer #2 (Remarks to the Author):

I appreciate the authors attention to my earlier concerns. Ultimately, I would not have found the applied revisions sufficient for acceptance had they not included the synteny analysis of an independent assembly of the focal species' genome. I fear that as more and more researchers are able to generate chromosomal level assemblies with long-read data alone, we will find that Hi-C scaffolding has led to numerous contig ordering errors. Luckily, in the present case, such errors appear to be absent. As such, I believe the present draft is satisfactory for publication.

Answer: We are grateful to the reviewer for their insightful comments.

Reviewer #3 (Remarks to the Author):

The answer to my comment on the lack of novelty of this manuscript is not satisfactory. Obviously, some novel findings are described but they do not represent a breakthrough as some of the ones previously published in top journals by the same group. In my opinion, having published on top journals should not guarantee future publications on top journals. I keep thinking that a more specialized journal would have been more appropriate for this manuscript. However, I must take note of the opinion of the other two reviewers and of the fact that the paper was sent out to referees. Most other comments and changes inserted in the new version of the manuscript are appropriate.

Answer: Although we recognize the reviewer's viewpoint on the importance of our findings, we maintain that our results have implications that extend beyond the interests of a narrowly defined audience.

An overarching objective of our research is to establish high-quality genomics for holocentric organisms, given that holocentricity represents a substantial disruption of the conventional genome structure. This allows us to study genomic features and processes in unique circumstances. The manuscript under consideration demonstrates that the holocentromeres of the woodrush, *Luzula sylvatica*, are not only a compelling subject for elucidating the transition to holocentricity but, as our findings reveal, offer a distinctive model for investigating centromere identity and the mechanisms that shape the genome landscape in general.

Moreover, the title and abstract of the manuscript have been modified to more accurately reflect our primary findings.

In the case of monocentrics, the study of centromere identity and evolution is frequently constrained by the limitations of either comparative studies or the examination of individual neocentromeres and evolutionary new centromeres. In contrast, our research has identified a multitude of apparent instances of dynamically evolving centromeric units within the genome of *L. sylvatica*. This has enabled us to describe the competing centromeric satellite repeats and their potential centromere stabilization role, as well as to identify neocentromere-like units with an unexpected epigenetic signature.

This illustrates the potential contribution of holocentric plants to the broader field of genomics, which motivates us to disseminate our findings to the broader audience of *Nature Communications*.

The only point requiring further clarification is the description of the new Supplementary Figure S11. The Figure legend is wrong. Specific description of panels a, b, c, d and e should be added. In particular an explanation of panel e should be better detailed. Are entire chromosomes or portions of chromosomes depicted? What is the scale? Chromosome numbers for *J. effusus* are missing. The description of this Figure in the text should also be clarified.

Answer: We are grateful to the reviewer for bringing this to our attention. We have corrected the description of Suppl. Fig. 12 (former Figure S11) and expanded the description of the panels based on your suggestions. We have also improved the inclusion of the Suppl. Fig. 11 to the main manuscript by adding a description of the procedure in the methodology section.

Reviewer #4 (Remarks to the Author):

Answer: We thank the reviewer for participating in this training program and for contributing to the peer review of our manuscript.